# HUBER ADDITIVE MODELS FOR NON-STATIONARY TIME SERIES ANALYSIS

**Yingjie Wang**[1,2], **Xianrui Zhong**[3], **Fengxiang He**[2,*]**, Hong Chen**[4,*] **& Dacheng Tao**[2]
[1]College of Informatics, Huazhong Agricultural University, China
[2]JD Explore Academy, JD.com Inc, China
[3]Department of Computer Science, University of Illinoist at Urbana-Champaign, USA
[4]College of Science, Huazhong Agricultural University, China
`yingjiewang1201@gmail.com, xzhong23@illinois.edu,`
`fengxiang.f.he@gmail.com, chenh@mail.hzau.edu.cn,`
`dacheng.tao@gmail.com`

## ABSTRACT

Sparse additive models have shown promising flexibility and interpretability in processing time series data. However, existing methods usually assume the time series data to be stationary and the innovation is sampled from a Gaussian distribution. Both assumptions are too stringent for heavy-tailed and non-stationary time series data that frequently arise in practice, such as finance and medical fields. To address these problems, we propose an adaptive sparse Huber additive model for robust forecasting in both non-Gaussian data and (non)stationary data. In theory, the generalization bounds of our estimator are established for both stationary and non-stationary time series data, which are independent of the widely used mixing conditions in learning theory of dependent observations. Moreover, the error bound for non-stationary time series contains a discrepancy measure for the shifts of the data distributions over time. Such a discrepancy measure can be estimated empirically and used as a penalty in our method. Experimental results on both synthetic and real-world benchmark datasets validate the effectiveness of the proposed method. The code is available at `https://github.com/xianruizhong/SpHAM`.

## 1 INTRODUCTION

Additive model has become one of the most powerful tools for time series analysis due to the exemplary monograph (Stone, 1985; Hastie & Tibshirani, 1990) and companion software (Chambers & Hastie, 1992). For the past two decades, the growing importance of algorithmic flexibility and interpretability motivates the development of various additive models along with theoretical explorations (Huang & Yang, 2004; Wang & Yang, 2007; Chu & Glymour, 2008; Song & Yang, 2010; Yang et al., 2018; Chen et al., 2017; Liu et al., 2020; Chen et al., 2021a) and practical applications (Dominici et al., 2002; Wang & Brown, 2011; Ravindra et al., 2019; Bussmann et al., 2020; Wang et al., 2020). Although these aforementioned works have shown promising behaviours, the proposed methods in these works require some stringent assumptions on the stochastic process, e.g., various mixing conditions (Doukhan, 1994), stationary distribution and Gaussian innovation.

A number of attempts have been made to relax such stringent assumptions. For the purpose of dealing with non-Gaussian innovation, Qiu et al. (2015) develop an elliptical vector autoregressive model for estimating heavy-tailed stationary processes with parametric convergence analysis that reduces the influence of non-Gaussian innovation. Moreover, under stationarity and decaying $\beta$-mixing condition, Wong et al. (2020) derive nonasymptotic estimation error of lasso without assuming special parametric form of the data generating process. Stationarity and various mixing conditions are commonly adopted in many previous studies, see, e.g., (Mohri & Rostamizadeh, 2009; 2010; Kock & Callot, 2015; Wong et al., 2020). In practice, the mixing and stationary conditions are too stringent and not always valid (Baillie, 1996; Kuznetsov & Mohri, 2020a). To relax the mixing and stationary conditions, Adams & Nobel (2010) prove asymptotic guarantees for stationary ergodic

---

*Corresponding authors.

sequences. Agarwal & Duchi (2013) establish generalization bounds for asymptotically stationary (mixing) processes in the case of stable on-line learning algorithms. Kuznetsov & Mohri (2014) establish learning guarantees for fully non-stationary and mixing processes. Recently, Kuznetsov & Mohri (2020a) provide data-dependent generalization risk bounds for non-stationary and non-mixing stochastic processes.

However, the exploration of additive models for both robust and non-stationary time series forecasting is still very limited. In this paper, we propose a class of sparse Huber additive models with theoretical guarantees. Our main contributions are summarized as follows:

- *Algorithm design and theoretical guarantees*: In Section 3.1, we first propose a novel sparse Huber additive model (SpHAM) with Huber loss, sparsity-inducing $\ell_{2,1}$-norm regularizer and additive data-dependent hypothesis space. This proposed method works for stationary time series and can achieve robust forecasting and satisfactory inference (e.g., Granger causal discovery) simultaneously. In theory, Section 3.2 establishes the upper bound of the function approximation error of SpHAM by developing error decomposition technique (Wu et al., 2006; Chen et al., 2021b) and employing sequential Rademacher complexity (Rakhlin et al., 2010; 2015; Kuznetsov & Mohri, 2020a). With properly selected scale parameters, the theoretical findings indicate that: a) For stationary time series, the function approximation consistency with convergence rate $O(T^{-\frac{1}{2}})$ can be pursued, even if the innovation is non-Gaussian distribution (see Theorem 1 and Corollary 1 for more details). Moreover, this consistency analysis appears to be novel because no explicit data dependence assumptions are not imposed here, e.g., various mixing conditions used in (Doukhan, 1994; Mohri & Rostamizadeh, 2010; Zou et al., 2009; Wong et al., 2020); b) For non-stationary time series, the function approximation error is bounded by a discrepancy measure, which characterizes the drifts of the data distributions along with time (see Theorem 2 for more details). By penalizing such a discrepancy measure, Section 3.3 further proposes an adaptive SpHAM for non-stationary time series and provides its theoretical upper bound correspondingly (see Theorem 3 for more details).
- *Optimization and empirical evaluations*: The proposed SpHAM and adaptive SpHAM can be implemented efficiently by Fast Iterative Shrinkage-Thresholding Algorithm (FISTA) (Beck & Teboulle, 2009). Experimental results on both synthetic and real-world benchmark CauseMe (Runge et al., 2019) validate the effectiveness of the proposed method.

**Related works**: There exist some interesting studies towards sparse additive model for time series analysis from both theoretical and practical viewpoints; see, e.g., Chu & Glymour (2008); Song & Yang (2010); Yang et al. (2018). Although they show promising interpretability and modeling capacity, all of them require Gaussian innovation, stationarity and various mixing conditions. Moreover, these assumptions are not always valid in heavy-tailed and non-stationary time series data. In contrast to these previous additive models, we are seeking to formulate our method and investigate its asymptotic properties without resorting to such strict assumptions.

Robust forecasting is a vibrant area of research in time series analysis (Qiu et al., 2015; Wong et al., 2020). As one of the triumphs and milestones of robust statistics, the success stories of Huber algorithms are mainly on robust prediction tasks with i.i.d datasets (Huber, 1964; Huber & Ronchetti, 2009; Loh, 2017; Feng & Wu, 2020), and their extensions to time series prediction are fairly sparse. To our best knowledge, this is the first work that considers Huber additive models for non-stationary and dependent time series data.

Our theoretical analyses are inspired from the successful usage of sequential Rademacher complexity in Kuznetsov & Mohri (2020b;a). The generalization risk bounds are established in their works for a general scenario of non-stationary and non-mixing stochastic processes. However, it should be mentioned that we are chiefly concerned about the function approximation analysis, which is essentially different from theirs and is very crucial for Huber algorithms, since the convergence of generalization risk cannot imply the convergence of function approximation (Sun et al., 2019; Feng & Wu, 2020; He & Tao, 2020). Moreover, the development of analysis techniques (e.g., error decomposition and sequential Rademacher complexity) for assessing our method may shed light on other robust models for nonstationary time series analysis.

To better highlight the novelty of our method, Table 1 summarizes the algorithmic properties of our method and other related works, e.g., Sparse additive models for time series (TS-SpAM) (Yang et al.,

Table 1: The properties of related methods

|  | TS-SpAM | DBF | R-Dantzig | Ours |
|---|---|---|---|---|
| Hypothesis space | Additive Spline-based | Kernel-based | Linear | Additive Kernel-based |
| Loss | Squared | Squared | Quantile-based | Huber |
| Mixing condition | Yes | No | Yes | No |
| Stationarity | Yes | No | Yes | No |
| Robustness | No | No | Yes | Yes |
| Sparsity | Yes | No | Yes | Yes |

2018), Discrepancy-based Forecasting (DBF) (Kuznetsov & Mohri, 2020a) and Robust Dantzig-selector-type estimator (R-Dantzig) (Qiu et al., 2015).

The remainder of this paper is organized as follows. Section 2 recalls the background of additive models. Section 3 mainly provides our methods and theoretical guarantees. We provide empirical evaluations in Section 4. Finally, Section 5 concludes this paper. The source code package is available at https://github.com/xianruizhong/SpHAM.

## 2 PRELIMINARY

Let $\{Z^t\}_{t=-\infty}^{\infty}$ be a stochastic time series with time index $t$, where variable $Z^t = (X^t, Y^t)$ takes values in the compact input space $\mathcal{X} \subset \mathbb{R}^p$ and the output space $\mathcal{Y} \subset \mathbb{R}$. We consider a common nonparametric model

$$Y^t = f^*(X^t) + \varepsilon^t, \ \ \mathbb{E}(\varepsilon^t) = 0, \tag{1}$$

where $f^*(\cdot)$ is the ground truth function, and the innovation $\varepsilon^t$ is i.i.d. across time $t \in \mathbb{Z}$. For the sake of simplicity, we denote $\rho^t$ and $\rho_{\mathcal{X}}^t$ as the jointed distribution of $(X^t, Y^t)$ and the corresponding marginal distribution with respect to $X^t$, respectively. This setup actually covers a large number of scenarios commonly used in practice. For instance, the case $X^t$ that contains $p$ lagged values of $Y^t$ (e.g., $X^t = Y^{(t-p)} \times \cdots \times Y^{(t-1)}$) corresponds to the $p$-order autoregressive models. Moreover, this case can be viewed as a vector autoregressive model in the sense that input $X^t$ includes the historical information of multiple variables.

Although such a nonparametric model (1) makes very few assumptions on data generation, the related nonparametric algorithms suffer so-called "curse of dimensionality", see Fan & Gijbels (1996) for further discussion. An effective strategy for solving this problem is additive model. Usually, the additive structure is obtained by decomposing the input space $\mathcal{X} \in \mathbb{R}^p$ into $\mathcal{X} = \mathcal{X}_1 \times \dots \times \mathcal{X}_p$. Under the assumption that the ground truth admits an additive structure $f^* = \sum_{j=1}^p f_j^*$, the additive model can be defined by

$$Y^t = f_1^*(X_1^t) + \cdots + f_p^*(X_p^t) + \varepsilon^t, \tag{2}$$

where each component $f_j^* : \mathcal{X}_j \to \mathbb{R}$ is a smooth function. In linear time series analysis, a weak stationarity condition (i.e., the first two moments of time series are time invariant) is preferred (Han et al., 2015; Qiu et al., 2015). In contrast, strict stationarity is primarily used to analyse the nonlinear time series (Fan & Yao, 2005).

**Definition 1.** *A stochastic process $\{Z^t\}_{t=-\infty}^{\infty}$ is strictly stationary if $(Z^1, ..., Z^t)$ and $(Z^{1+k}, ..., Z^{t+k})$ have the same joint distributions for any $t \in \mathbb{Z}$ and $k \in \mathbb{Z}$.*

Note that, if not otherwise stated, the stationarity in this paper refers to strict stationarity. Suppose that we are given $T$ size time series data $\{(x^t, y^t)\}_{t=1}^T \in \mathcal{Z}^T$ which are drawn from an additive data-generating model (2). Under stationarity condition and zero-mean Gaussian innovation with finite variance, the widely used methods that learn the ground truth $f^*$ usually integrate squared loss and a smoothness- or sparsity-inducing regularizer $\Omega(\cdot)$ into a structural risk minimization scheme:

$$\min_{f \in \mathcal{H}} \sum_{t=1}^T (y^t - \sum_{j=1}^p f_j(x_j^t))^2 + \Omega(f),$$

where $\mathcal{H} := \{f_1 + ... + f_p : f_j \in \mathcal{H}_j, j = 1, ..., p\}$ is an additive hypothesis space. Commonly, each subspace $\mathcal{H}_j$ could be reproducing kernel Hilbert space (Chen et al., 2017; Kandasamy & Yu, 2016; Raskutti et al., 2012), the orthogonal basis inducing space (Ravikumar et al., 2009; Meier et al., 2009; Yang et al., 2018), or the composite function space with the neural network as a typical (Agarwal et al., 2020; Bussmann et al., 2020).

However, when facing heavy-tailed innovation, these methods may have degraded performance due to the amplification of the squared loss to large residuals. As one commonly used statistic in robust learning community (Peng et al., 2019; Wang et al., 2017b; Feng & Wu, 2020), Huber loss is defined as

$$\ell_\sigma(f(x^t) - y^t) = \begin{cases} (f(x^t) - y^t)^2, & \text{if } |f(x^t) - y^t| < \sigma \\ 2\sigma|f(x^t) - y^t| - \sigma^2, & \text{if } |f(x^t) - y^t| \geq \sigma, \end{cases} \tag{3}$$

where $\sigma$ is a positive hyper-parameter. Note that in the previous studies (Huber & Ronchetti, 2009; Loh, 2017), the hyper-parameter $\sigma$ is set to be fixed according to the 95% asymptotic efficiency rule. However, Huber regression with a fixed scale parameter may not be able to learn the ground truth when the noise is asymmetric, as argued recently in Feng & Wu (2020); Sun et al. (2019). In this paper, we choose the scale parameter $\sigma$ by relating it to the moment condition of the noise distribution and the sample size so that the resulting regression estimator can asymptotically converge to the ground truth function.

# 3 METHOD

## 3.1 SPARSE HUBER ADDITIVE MODELS

In this paper, we choose reproducing kernel Hilbert space (RKHS) $\mathcal{H}_{K_j}, j = 1, .., p$, to form the the additive hypothesis space $\mathcal{H}$, where each $\mathcal{H}_{K_j}$ is associated with a symmetric and positive semi-definite Mercer kernel $K_j : \mathcal{X}_j \times \mathcal{X}_j \to \mathbb{R}$. An additive RKHS is defined as

$$\mathcal{H}_K = \{f_1 + ... + f_p : f_j \in \mathcal{H}_{K_j}, j = 1, ..., p\} \tag{4}$$

with kernel norm $\|f\|_K^2 = \inf\{\|f\|_{K_1}^2 + ... + \|f\|_{K_p}^2\}$. By integrating the Huber loss (3), additive RKHS and kernel norm inducing regularizer into a Tikhonov regularization scheme, the regularized Huber additive model with kernel-norm can be formulated as

$$\hat{f}_\eta = \sum_{j=1}^p \hat{f}_{\eta,j} = \underset{f=\sum_{j=1}^p f_j, f_j \in \mathcal{H}_{K_j}}{\arg\min} \{\sum_{t=1}^T \ell_\sigma(y^t - \sum_{j=1}^p f_j(x_j^t)) + \eta \sum_{j=1}^p \tau_j \|f_j\|_{K_j}^2\}, \tag{5}$$

where $\eta$ is positive regularization parameter and $\tau_j$ is the weight for $j$-th kernel norm. The representer theorem (Wahba, 1990) ensures that $\hat{f}_\eta$ can be represented as

$$\hat{f}_\eta = \sum_{j=1}^p \sum_{t=1}^T \alpha_{tj}^\eta K_j(x_j^t, \cdot), \ \alpha_{tj}^\eta \in \mathbb{R}.$$

To offer the method with sparsity, we consider the following sparsity-inducing penalty

$$\Omega(f) := \inf\{\sum_{j=1}^p \tau_j \|\alpha_j\|_2 : f = \sum_{j=1}^p \sum_{t=1}^T \alpha_{tj} K_j(x_j^t, \cdot)\}.$$

Let $\mathcal{H}_Z$ be an additive data dependent hypothesis space defined by

$$\mathcal{H}_Z = \{f = \sum_{j=1}^p \sum_{t=1}^T \alpha_{tj} K_j(x_j^t, \cdot) : \alpha_{tj} \in \mathbb{R}\}.$$

Then the SpHAM can be formulated as

$$\hat{f} = \underset{f \in \mathcal{H}_Z}{\arg\min}\{\frac{1}{T}\sum_{t=1}^T \ell_\sigma(y^t - \sum_{j=1}^p f_j(x_j^t)) + \lambda\Omega(f)\}.$$

Denote $\mathbf{K}_j^t = (K_j(x_j^1, x_j^t), ..., K_j(x_j^T, x_j^t))' \in \mathbb{R}^T$, $\alpha = (\alpha_1', ..., \alpha_p')' \in \mathbb{R}^{Tp}$ and $\alpha_j = (\alpha_{1j}, ..., \alpha_{Tj})' \in \mathbb{R}^T$, where $\alpha_j'$ here refers to the transpose of $\alpha_j$ for avoiding the confusion. The SpHAM can be represented as

$$\hat{f} = \sum_{j=1}^{p} \sum_{t=1}^{T} \alpha_{tj}^{\lambda} K_j(x_j^t, \cdot) \tag{6}$$

with

$$\alpha^{\lambda} = \underset{\alpha_j \in \mathbb{R}^T, j=1,...,p}{\arg\min} \{\frac{1}{T} \sum_{t=1}^{T} \ell_{\sigma}(y^t - \sum_{j=1}^{p} (\mathbf{K}_j^t)' \alpha_j) + \lambda \sum_{j=1}^{p} \tau_j \|\alpha_j\|_2\}. \tag{7}$$

The optimization problem (7) can be solved by Fast Iterative Shrinkage-Thresholding Algorithm (FISTA) (Beck & Teboulle, 2009). We provide the detailed optimization procedure in Appendix E.

**Remark 1.** *Inspired by the studies on group additive models (Yin et al., 2012; Pan & Zhu, 2017; Chen et al., 2017), the modeling capacity of our proposed method can be improved by embedding the (Hierarchical) group structure. For instance, group SpHAM can be easily formulated by replacing the direct decomposition $\{\mathcal{X}_j\}_{j=1}^p, \mathcal{X}_j \in \mathbb{R}$ with the subgroups decomposition $\{\mathcal{X}_d\}_{d=1}^D$, where each $\mathcal{X}_d \subset \mathcal{X}$ may cover more than one variable.*

## 3.2 ASYMPTOTIC THEORY ANALYSIS

In this section, we provide the theoretical analyses of SpHAM, including the following:

- The function approximation errors of SpHAM for stationary time series data (Theorem 1; Section 3.2.1) and non-stationary time series data (Theorem 2; Section 3.2.2);
- An adaptive SpHAM (inspired by above theoretical findings) and its function approximation error for non-stationary time series (See Theorem 3; Section 3.2.3)

Due to space limitation, the high-level outline and detailed proofs are provided in the Appendix A-D.

### 3.2.1 FUNCTION APPROXIMATION ANALYSIS FOR STATIONARY TIME SERIES

Through this paper, the marginal distribution w.r.t $Y^t$ is assumed to be almost everywhere supported on $[-M, M]$ for some $M \geq 0$.

**Assumption 1.** *Assume that $|Y^t|$, $\forall t \in \mathbb{Z}$, is bounded and there exists a constant $c > 0$ such that $\mathbb{E}|Y^t|^{1+c} < \infty$, $\forall t \in \mathbb{Z}$.*

The moment condition in Assumption 1 is rather weak in the sense that the response variable $Y^t$ possesses infinite variance. The same condition also applies to the distributions of the innovation $\varepsilon^t$, implying that heavy-tailed innovation is allowed.

**Assumption 2.** *Let $\kappa = \sup_{x \in \mathcal{X}} \sqrt{K_j(x, x)} < \infty, \forall j = 1, ..., p$.*

Assumption 2 only requires that the kernel is bounded under compact $\mathcal{X}$, which holds for all Mercer kernels (e.g., Gaussian kernel) and has been used in many learning theory literatures; see, e.g., (Wu et al., 2006; Steinwart & Christmann, 2008; Wu & Zhou, 2008).

The following definitions are needed for Assumption 3. For any $j = 1, ..., p$, we define a kernel integral operator $L_{K_j, T+1} : L_2(\rho_{\mathcal{X}_j}^{T+1}) \to L_2(\rho_{\mathcal{X}_j}^{T+1})$ associated with the kernel $K_j$ by

$$L_{K_j, T+1}(f)(x_j^{T+1}) = \int_{\mathcal{X}_j} K_j(x_j^{T+1}, u_j) f(u_j) d\rho_{\mathcal{X}_j}^{T+1}(u_j).$$

Note that $L_{K_j, T+1}$ is a compact and positive operator on $L_2(\rho_{\mathcal{X}_j}^{T+1})$. According to Mercer theorem, we can find the corresponding normalized eigenpairs $\{(\zeta_i^j, \psi_i^j)\}_{i \geq 1}$ such that $\{\psi_i^j\}_{i \geq 1}$ is an orthonormal basis of $L_2(\rho_{\mathcal{X}}^{T+1})$ and $\zeta_i^j \to 0$ as $i \to \infty$. Then for given $r > 0$, we defined the $r$-th power $L_{K_j, T+1}^r$ by

$$L_{K_j, T+1}^r(\sum_{i \geq 1} \beta_i^j \psi_i^j) = \sum_{i \geq 1} \beta_i^j (\zeta_i^j)^r \psi_i^j.$$

**Assumption 3.** *We assume that $f_j^* : \mathcal{X}_j \to \mathbb{R}, \forall j = 1, ..., p$ is a function of the form $f_j^* = L_{K_j, T+1}^r(g_j^*), \forall r \in (0, \frac{1}{2}]$ with some $g_j^* \in L_2(\rho_{\mathcal{X}_j}^{T+1}), \forall h \in \mathbb{Z}$.*

Assumption 3 is a natural extension from i.i.d setting (Assumption 1 in (Wu et al., 2006; Christmann & Zhou, 2016; Chen & Wang, 2018)) to non-i.i.d time series. Indeed, this assumption stands in most practical cases. For instance, if $r = 0.5$, the ground truth function $f^*$ needs to be a real-valued function in the RKHS $\mathcal{H}_K$. The RKHS admits a large class of bounded and continuous real-valued functions that are in Hilbert space. Moreover, this assumption has been widely used in learning theory; please refer to (Smale & Zhou, 2003; Cucker & Zhou, 2007) for more discussions.

**Theorem 1.** *Suppose that the process $\{Z^t\}_{t=-\infty}^{\infty}$ is stationary. Let Assumptions 1-3 be true. By taking $\sigma = T^m, \eta = T^{-\frac{1}{4r}}$ and $\lambda = T^{-\frac{1}{4r}-m}$, we have for any $0 < \delta < 1$,*

$$\|\hat{f} - f^*\|_{L_2(\rho_{\mathcal{X}}^{T+1})}^2 \leq \widetilde{C} \log(1/\delta) T^{\Psi(m,c,r)}$$

*with confidence at least $1 - \delta$, where $\widetilde{C}$ is a positive constant independently of $T, \lambda, \eta, \delta$ and $\sigma$ and*

$$\Psi(m, c, r) = \begin{cases} \max\{-\frac{1}{2}, m-1, -cm + \frac{1}{4r} + m - 1\}, & \text{if } m \leq 1 - \frac{1}{4r} \\ \max\{-\frac{1}{2}, m-1, -cm + \frac{1}{2r} + 2m - 2\}, & \text{if } m > 1 - \frac{1}{4r}. \end{cases}$$

**Remark 2.** *In the stationary case, our bound appears to be novel for the following reasons: a) the result is completely independent of the various mixing conditions which are widely used in the theory analysis of non-i.i.d dependent time series (Mohri & Rostamizadeh, 2009; Yang et al., 2018; Mohri & Rostamizadeh, 2010; Guo & Shi, 2011; Qiu et al., 2015); b) compared with Yang et al. (2018), the innovation assumption is rather weak in the sense that the innovation possesses infinite variance and thus admits a heavy-tailed distribution.*

**Corollary 1.** *Suppose that the process $\{Z^t\}_{t=-\infty}^{\infty}$ is stationary. Let all the conditions in Theorem 1 be true. We then have for any $0 < \delta < 1$*

$$\|\hat{f} - f^*\|_{L_2(\rho_{\mathcal{X}}^{T+1})}^2 \leq \widetilde{C} \log(1/\delta) T^{\Psi(m,c)}$$

*with confidence at least $1 - \delta$, where*

$$\Psi(m, c) = \begin{cases} \max\{-\frac{1}{2}, -cm + m - \frac{1}{2}\}, & \text{if } m \leq \frac{1}{2} \\ \max\{m-1, -cm + +2m - 1\}, & \text{if } m > \frac{1}{2}. \end{cases}$$

Figure 1 summaries the convergence rates in Corollary 1 by taking different $\sigma$ and $c$. If the innovation is Gaussian distribution with finite variance (i.e., Assumption 1 holds for any $c > 1$), one can arbitrarily select a $\sigma = T^m$ $(0 < m < \frac{1}{2})$ to obtain convergence rates $O(T^{-\frac{1}{2}})$. Moreover, we can see that the convergence rate will decrease as $m$ increases. Combined with the conclusion in Lemma 1 (i.e., the equivalence relation between Huber loss based empirical risk and MSE as $\sigma \to \infty$), it indicates that $\sigma$ indeed plays a trade-off role between algorithmic robustness and variance-reduction. For the weak moment condition (e.g., $0 < c < 1$), one may get slower convergence rates (e.g., $O(T^{(1-c)m-\frac{1}{2}})$ or

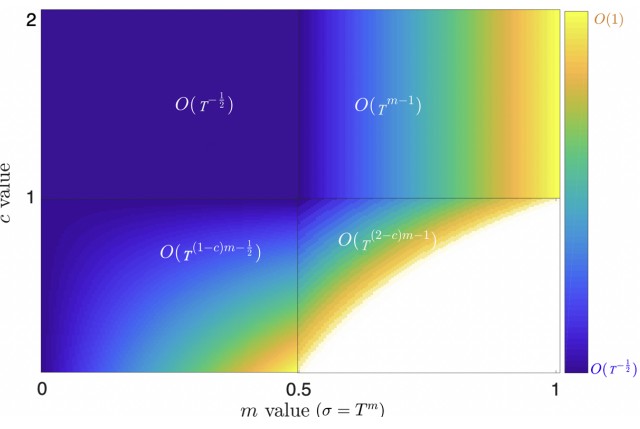

Figure 1: The convergence rates under different $\sigma$ and $c$.

$O(T^{(2-c)m-1})$), which also coincides with our intuitive understanding that small $\sigma$ may be conducive to robust forecasting. Note that our method will not converge when $\sigma$ and $c$ are both located in the white area in Figure 1.

As a comparison, due to the non-robustness of the squared loss, the most existing convergence rates are established under the assumption that the innovation is Gaussian distribution with finite variance (see, e.g., Yang et al. (2018); Han et al. (2015); Wang & Yang (2007); Kock & Callot (2015)). However, from Theorem 1 and Corollary 1, the asymptotic convergence of SpHAM can be obtained under weaker moment condition, which verifies the robustness of our method. Recall the learning rate $O(T^{\frac{-2d}{2d+1}})$ derived in Yang et al. (2018), where $d$ is the order of smoothness of the component function $f_j, j = 1, ..., p$. Relatively slow convergence rate $O(T^{-\frac{1}{2}})$ we obtained indicates the sacrifice for the absence of mixing condition.

### 3.2.2 Function approximation analysis for non-stationary time series

In nonstationary time series setting, different $Z^t$s may follow different distributions. Thus c. For given weights $\{s_t\}_{t=1}^T$, we define the estimator $\hat{f}^\mathbf{s} = \sum_{j=1}^p \hat{f}_j^\mathbf{s}$ as the minimizer of the following weighted objective

$$\mathcal{E}_\lambda^\mathbf{s}(\hat{f}^\mathbf{s}) := \min_{f \in \mathcal{H}_Z}\{\sum_{t=1}^T s_t\ell_\sigma(y^t - \sum_{j=1}^p f_j(x_j^t)) + \lambda\Omega(f)\}. \tag{8}$$

Next, we introduce a discrepancy measure to characterize the discrepancy between the target distribution and the distributions of observations (Kuznetsov & Mohri, 2020a).

**Definition 2.** *For any $f \in \mathcal{H}_Z$, the discrepancy measure with respect to Huber loss is defined as*

$$\text{disc}(\mathbf{s}) := \sup_{f \in \mathcal{H}_Z}\left\{\mathbb{E}\ell_\sigma(f(x^{T+1}) - y^{T+1}) - \sum_{t=1}^T s_t\mathbb{E}\ell_\sigma(f(x^t) - y^t)\right\}.$$

The discrepancy measures the non-stationarity of the stochastic process $\{Z^t\}_{t=-\infty}^\infty$ with respect to both the loss function $\ell_\sigma$ and the hypothesis set $\mathcal{H}_Z$.

**Theorem 2.** *Let Assumptions 1-3 be true. We assume that $f_j^* \in \mathcal{H}_{K_j}, \forall j = 1, ..., p$. By taking $\sigma = T^{\frac{1}{2c}}$, $\lambda = T^{-1}$ and $\eta = T^{-\frac{1}{2}}$, we have for any $0 < \delta < 1$*

$$\|\hat{f}^\mathbf{s} - f^*\|_{L_2(\rho_\mathcal{X}^{T+1})}^2 \leq \text{disc}(\mathbf{s}) + \mathcal{E}_\lambda^\mathbf{s}(\hat{f}^\mathbf{s}) + \widetilde{C}_1\|\mathbf{s}\|_2 T^{\frac{1}{2c}} + \widetilde{C}_2\log(1/\delta)T^{-\frac{1}{2}}$$

*with confidence at least $1 - \delta$, where $\widetilde{C}_1, \widetilde{C}_2$ are two positive constants independently of $T, \lambda, \eta, \delta$ and $\sigma$.*

Note that there are several existing studies towards analyzing nonstationary and non-mixing time series, see, e.g., Kuznetsov & Mohri (2020a). Different from them, the error bound we derived is with respect to the function approximation rather than generalization risk, which is very crucial for Huber regression problem, since the convergence of generalization risk cannot imply the convergence of function approximation (Sun et al., 2019; Feng & Wu, 2020).

### 3.2.3 Adaptive sparse Huber additive model for nonstationary time series

Theorem 2 illustrates that we shall minimize the following optimization problem for non-stationary time series forecasting:

$$\min_{f \in \mathcal{H}_Z}\{\sum_{t=1}^T s_t\ell_\sigma(y_t - \sum_{j=1}^p f_j(x_j^t)) + \text{disc}(\mathbf{s}) + \lambda_1\Omega(f) + \lambda_2 T^{\frac{1}{2c}}\|\mathbf{s}\|_2\},$$

where $\lambda_1$ and $\lambda_2$ are two positive regularization parameters, and $c$ is a positive constant introduced in Assumption 1. Although the discrepancy measure disc$(\mathbf{s})$ is crucial for such an optimization problem, we cannot obtain its exact value since we do not have access to the distributions of $Z^t, t \in \mathbb{Z}$. Hence, we need to estimate the approximated discrepancy from given data. Inspired by Kuznetsov & Mohri (2020a), one natural and necessary assumption is that there exists an underlying representation relationship between distribution $\rho^{T+1}$ and distributions $\rho^t, t = 1, ..., T$.

**Assumption 4.** *Denote by a probability set* $\mathbf{q}^* = \{q_t^*\}_{t=1}^T$ *with* $\sum_{t=1}^T q_t^* = 1$. *We assume that the following term is sufficiently small:*

$$\text{disc}(\mathbf{q}^*) := \sup_{f \in \mathcal{H}_Z} [\mathbb{E}\ell_\sigma(f(x^{T+1}) - y^{T+1}) - \sum_{t=1}^T q_t \mathbb{E}\ell_\sigma(f(x^t) - y^t)].$$

Note that the priori $\mathbf{q}^*$ can be any distribution, which shall be given empirically according to the trend of the time series. For instance, in a particular scenario of the distribution $\rho^{T+1}$ does not change drastically compared with the distributions $\rho^t, t = 1, ..., T$, Kuznetsov & Mohri (2020a) have proven that the Assumption 4 holds if the $\mathbf{s}^*$ is an uniform distribution over last $l > 0$ observations, where $l$ is a hyper-parameter that can be tuned in practical applications.

**Theorem 3.** *Let Assumption 4 and the conditions in Theorem 2 be true. We have for any* $0 < \delta < 1$

$$\|\hat{f}^{\mathbf{s}} - f^*\|_{L_2(\rho_{\mathcal{X}}^{T+1})}^2 \leq \sup_{f \in \mathcal{H}_Z} \sum_{t=1}^T (q_t^* - s_t)\ell_\sigma(f(x^t) - y^t) + \mathcal{E}_\lambda^{\mathbf{s}}(\hat{f}^{\mathbf{s}})$$

$$+ \widetilde{C}_1 \log(1/\delta)(\|\mathbf{q}^* - \mathbf{s}\|_2 + \|\mathbf{s}\|_2 T^{\frac{1}{2c}}) + \widetilde{C}_2 \log(1/\delta) T^{-\frac{1}{2}}$$

*with confidence at least* $1 - \delta$, *where* $\widetilde{C}_1, \widetilde{C}_2$ *are two constants independently of* $T, \lambda, \eta, \delta$ *and* $\sigma$.

Finally, the optimization problem of adaptive SpHAM can be formulated as following two stages:

**Step A: finding the weight** $\hat{\mathbf{s}}$:

$$\hat{\mathbf{s}} = \arg\min_{\mathbf{s}} \{ \sup_{f \in \mathcal{H}_Z} \sum_{t=1}^T (q_t^* - s_t)\ell_\sigma(f(x^t) - y^t) + \lambda_1\|\mathbf{q}^* - \mathbf{s}\|_2^2 + \lambda_2 T^{\frac{1}{2c}}\|\mathbf{s}\|_2^2 \} \tag{9}$$

**Step B: forecasting**:

$$\hat{f}^{\mathbf{s}}(x^{T+1}) = \sum_{j=1}^p \sum_{t=1}^T \alpha_{tj}^{\mathbf{s}} K(x_j^t, x_j^{T+1}),$$

where

$$\alpha^{\mathbf{s}} = \arg\min_{\alpha_j \in \mathbb{R}^T, j=1,...,p} \{ \sum_{t=1}^T \hat{s}_t \ell_\sigma(y^t - \sum_{j=1}^p (\mathbf{K}_j^t)'\alpha_j) + \lambda \sum_{j=1}^p \tau_j\|\alpha_j\|_2 \}. \tag{10}$$

The optimization problems (9) can be solved by standard gradient descent method. Similarly to the strategy for optimization problem (7), we use Fast Iterative Shrinkage-Thresholding Algorithm (FISTA) (Beck & Teboulle, 2009) for **Step B**. We provide the detailed procedure in Appendix E.

## 4 EXPERIMENT

This section validates the effectiveness of SpHAM and adaptive SpHAM. In all experiments, the Gaussian kernel $K_j(u, v) = \exp(-\frac{\|u-v\|_2^2}{2d^2})$, where $j = 1, ..., p$ and bandwidth $d > 0$, is employed for constructing the additive data dependent hypothesis space. Due to limited space, the evaluations on real-world data are provided in Appendix F. We consider two synthetic examples as below:

*Example A:* Inspired by (Kuznetsov & Mohri, 2020a), a stationary time series is generated according to the non-linear additive autoregressive model:

$$Y^t = \frac{3}{2}\sin(\frac{\pi}{2}Y^{t-2}) - \sin(\frac{\pi}{2}Y^{t-3}) + 2\varepsilon^t,$$

where the innovation $\varepsilon^t$s are i.i.d. drawn from Gaussian distribution $N(0, 0.5)$ and Student distribution with degree of freedom 2, respectively.

*Example B:* Inspired by (Kuznetsov & Mohri, 2020a), a time series with smooth drift is generated by

$$Y^t = \frac{t}{400}\sin(Y^{t-1}) + \frac{1}{2}\varepsilon^t,$$

where the distributions of noise $\varepsilon^t$ are the same as above.

*Hyper-parameter selection and evaluation criterions:* Recall that SpHAM algorithm requires three hyper-parameters: regularization parameter $\lambda$, bandwidth of kernel $d$ and Huber parameter $\sigma$. We set these parameters according to the suggestions in our Theorems. Based on the suggestion in Theorem 1-3, the selection of Huber parameter $\sigma$ is $\sigma = T^{\frac{1}{48}}$ and the regularization parameter is $\lambda = T^{-1}$. Moreover, we set the bandwidth $d = 0.5$ and tune $l \in \{100, 150, 200, ..., 350\}$. The evaluation criterions for forecasting used here contains Average Sample Error(ASE)$= \frac{1}{N}\sqrt{\sum_{t=1}^{N}(\hat{f}(x^t) - y^t)^2}$ and True Deviation (TD)$= \frac{1}{N}\sqrt{\sum_{t=1}^{N}(\hat{f}(x^t) - f^*(x^t))^2}$, where $N$ is the number of test samples.

For each example, we generate time series with $4000$ sample points. The samples at time $t = \{1500, 1501, ..., 1899\}$ are used as a training set, and the samples at next time $t = \{1900, ..., 1999\}$ are considered as the test data. The competitors include Simple Exponential Smoothing (SES) , TS-SpAM (Yang et al., 2018) and Vanilla Long Short-Term Memory (LSTM). The tuning parameters of other competing methods such as LSTM and SES, are chosen according to their original python packages. All the evaluations are repeated for $50$ times. The average results for *Examples A-B* are presented in Table 2.

From Table 2, the results on Example A verify that our SpHAM is competitive with other approaches based on square loss under Gaussian noise, and performs better in presence of heavy-tailed $t$ noise. Moreover, the results on Example B shows the promising performance of our adaptive SpHAM for non-stationary time series forecasting.

Table 2: The results on synthetic data.

|  | Methods | Gaussian noise | | Student noise | |
|---|---|---|---|---|---|
|  |  | ASE (Std) | TD (Std) | ASE (Std) | TD (Std) |
| Example A | SES | 0.114($\pm$.008) | 0.098($\pm$.005) | 0.493($\pm$.313) | 0.105($\pm$.009) |
|  | LSTM | **0.063**($\pm$.005) | **0.022**($\pm$.005) | 0.538($\pm$.342) | 0.219($\pm$.089) |
|  | TS-SpAM | 0.070($\pm$.007) | 0.035($\pm$.008) | 0.505($\pm$.332) | 0.131($\pm$.042) |
|  | SpHAM (ours) | 0.076($\pm$.005) | 0.042($\pm$.008) | **0.492**($\pm$.333) | **0.095**($\pm$.006) |
| Example B | SES | 0.435($\pm$.015) | 0.424($\pm$.015) | 0.402($\pm$.080) | 0.366($\pm$.017) |
|  | LSTM | 0.120($\pm$.125) | 0.114($\pm$.127) | 0.247($\pm$.125) | 0.181($\pm$.088) |
|  | TS-SpAM | 0.114($\pm$.022) | 0.105($\pm$.021) | 0.173($\pm$.125) | 0.104($\pm$.051) |
|  | SpHAM (ours) | 0.115($\pm$.028) | 0.112($\pm$.029) | 0.179($\pm$.116) | 0.096($\pm$.026) |
|  | Adaptive SpHAM (ours) | **0.102**($\pm$.026) | **0.096**($\pm$.026) | **0.172**($\pm$.118) | **0.085**($\pm$.024) |

## 5 CONCLUSION

We propose an adaptive sparse Huber additive model by integrating Huber loss and $\ell_{2,1}$-norm regularizer into an additive data dependent hypothesis space. We theoretically explore the asymptotic properties of our method for both non-Gaussian and (non)stationary time series. Experimental results on both synthetic and real-world data validate the effectiveness of the proposed method.

ACKNOWLEDGMENTS

This work is supported by the Major Science and Technology Innovation 2030 "New Generation Artificial Intelligence" key project (No. 2021ZD0111700) and the National Natural Science Foundation of China under Grant No. 12071166. We sincerely appreciate the anonymous ICLR reviewers for their helpful comments.

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

## A    PROOF SKETCH

This section sketches the proof focusing on the conceptual aspects. The full proofs are provided in Appendix B-D.

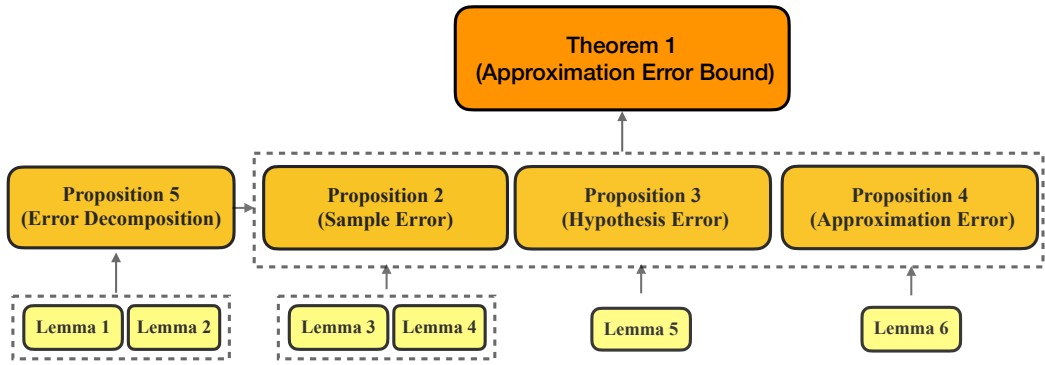

Figure 2: The important ingredients for Theorem 1.

Figure 2 summaries the important ingredients for the proof of Theorem 1. Lemma 1 describes the properties of $\hat{f}$ and Lemma 2 illustrates the relation between huber-based risk and MSE. Following these two lemmas and previous works (Wang et al., 2017a; Feng & Wu, 2020), we then decompose the function error into three important parts: sample error, hypothesis error and approximation error. To bound the sample error, we need to employ the sequential Rademacher complexity (Kuznetsov & Mohri, 2020a), which is the key to measuring the capacity of the data-dependent hypothesis space for non-i.i.d data. Lemmas 3-4 give an important concentration inequality and the upper bound of sequential Rademacher complexity, respectively. Finally, Theorem 1 is obtained by combining Propositions 2-4. The proof of Theorems 2 is similar to the above process, except that we develop a different error decomposition in Proposition 5. On the basis of Theorem 2, by applying an assumption on data distribution, we derive Theorem 3.

The full proofs of Theorems 1-3 are provided in Sections C-E, respectively.

## B    PROOF OF THEOREM 1

We first illustrate two key properties of $\hat{f}$.

**Lemma 1.** *Let Assumptions 1-2 be true. From the definition of $\hat{f}$ in Eq. (6), there hold*

$$\|\hat{f}\|_\infty \le M_{\hat{f}} := \frac{2\kappa^2 M^2}{\lambda T \min_{j=1,\ldots,p} \tau_j}.$$

*and*

$$\Omega(\hat{f}) = \sum_{j=1}^p \tau_j \|\alpha_j^\lambda\|_2 \le \kappa^{-2} M_{\hat{f}} = \frac{2M^2}{\lambda T \min_{j=1,\ldots,p} \tau_j},$$

*where $M$ is a positive constant such that $|Y^t| \le M, \forall t \in \mathbb{Z}$.*

*Proof.* Denote by $\mathcal{E}_T^\sigma(f) := \sum_{t=1}^T \ell_\sigma(f(x^{(t)}) - y^{(t)})$ for notational convenience. From the definition of $\hat{f}$, we know that

$$\mathcal{E}_T^\sigma(\hat{f}) + \lambda\Omega(\hat{f}) \le \mathcal{E}_T^\sigma(0).$$

It means that

$$\Omega(\hat{f}) = \sum_{j=1}^p \tau_j \|\alpha_j^\lambda\|_2 \le \lambda^{-1}\mathcal{E}_T^\sigma(0).$$

From the definition of Huber loss (3), we obtain

$$\mathcal{E}_T^\sigma(0) = \frac{1}{T}\sum_{t=1}^T \ell_\sigma(y^{(t)}) \le \sigma^2, \text{ if } |y^{(t)}| \le \sigma, \forall t = 1,...,T,$$

and

$$\mathcal{E}_T^\sigma(0) = \frac{1}{T}\sum_{t=1}^T \ell_\sigma(y^{(t)}) = 2\sigma|y^{(t)}| - \sigma^2 \le 2\sigma M - \sigma^2, \text{ if } |y^{(t)}| > \sigma, \forall t = 1,...,T.$$

Therefore,

$$\sum_{j=1}^p \tau_j \|\alpha_j^\lambda\|_2 \le \frac{2M^2}{\lambda T \min_{j=1,...,p}\tau_j}.$$

According to the property of RKHS, we conclude that

$$\|\hat{f}\|_\infty \le \kappa\|\hat{f}\|_K \le \kappa\sqrt{\sum_{j=1}^p \|\hat{f}_j\|_{K_j}^2} \le \kappa^2\sum_{j=1}^p \|\alpha_j^\lambda\|_2 \le \frac{2\kappa^2 M^2}{\lambda T \min_{j=1,...,p}\tau_j}.$$

This completes the proof by denoting $M_{\hat{f}} = \frac{2\kappa^2 M^2}{\lambda T \min_{j=1,...,p}\tau_j}$. $\qquad\square$

Let $\bar{\mathcal{H}}_Z$ be a data-dependent hypothesis space constrained by sparsity-inducing regularizer $\Omega(\cdot)$, i.e.,

$$\bar{\mathcal{H}}_Z = \{f(x) = \sum_{j=1}^p\sum_{i=1}^T \alpha_{ij} K_j(x_j^{(t)}, x_j) : \sum_{j=1}^p \tau_j\|\alpha_j\|_2 \le \frac{2M^2}{\lambda T \min_{j=1,..,p}\tau_j}\}.$$

For any $h \in \mathbb{Z}$, denote by $\mathcal{E}_{T+1}^\sigma(f) := \mathbb{E}\ell_\sigma(f(x^{T+1})-y^{T+1})$ and $\mathcal{E}_{T+1}(f) := \mathbb{E}(f(x^{T+1})-y^{T+1})^2$, respectively. In the following, we describe the relationship between $\mathcal{E}_{T+1}^\sigma(f)$ and $\mathcal{E}_{T+1}(f), \forall f \in \bar{\mathcal{H}}_Z$. Note that a similar proof is given in (Feng & Wu, 2020) for the i.i.d case.

**Lemma 2.** *Let Assumptions 1-2 be true. For any $h \in \mathbb{Z}$ and $f \in \bar{\mathcal{H}}_Z$, there holds*

$$\left|[\mathcal{E}_{T+1}^\sigma(f) - \mathcal{E}_{T+1}^\sigma(f^*)] - [\mathcal{E}_{T+1}(f) - \mathcal{E}_{T+1}(f^*)]\right| \le \frac{M_c}{\sigma^c},$$

*where both $c$ and $M_c = 2^{3+c}(M_{f^*} + M_{\hat{f}} + 1)^2\mathbb{E}(|Y^{T+1}|^{1+c})$ are positive constants.*

*Proof.* For any $\sigma > \max\{M + M_{\hat{f}}, 1\}$, we denote two events $I_Y$ and $II_Y$ as follows

$$I_{Y^{T+1}} = \{Y^{T+1} : |Y^{T+1}| \ge \sigma/2\}$$

and

$$II_{Y^{T+1}} = \{Y^{T+1} : |Y^{T+1}| < \sigma/2\}.$$

Then, for any $f \in \bar{\mathcal{H}}_Z$ and $\|f^*\|_\infty = M_{f^*} < \infty$, we have

$$\left|[\mathcal{E}_{T+1}^\sigma(f) - \mathcal{E}_{T+1}^\sigma(f^*)] - [\mathcal{E}_{T+1}(f) - \mathcal{E}_{T+1}(f^*)]\right|$$

$$= \left|\left[\int_{\mathcal{Z}}\ell_\sigma(f(x^{T+1})-y^{T+1})d\rho^{T+1} - \int_{\mathcal{Z}}\ell_\sigma(f^*(x^{T+1})-y^{T+1})d\rho^{T+1}\right] - \|f-f^*\|_{L_2(\rho_{\mathcal{X}}^{T+1})}^2\right|$$

$$= \left|\left[\int_{\mathcal{X}}\int_{\mathcal{Y}}\ell_\sigma(f(x^{T+1})-y^{T+1}) - \ell_\sigma(f^*(x^{T+1})-y^{T+1})d\rho_{\mathcal{Y}|\mathcal{X}}^{T+1}d\rho_{\mathcal{X}}^{T+1}\right] - \|f-f^*\|_{L_2(\rho_{\mathcal{X}}^{T+1})}^2\right|$$

$$\le \left|\int_{\mathcal{X}}\int_{I_{Y^{T+1}}\cup II_{Y^{T+1}}}\ell_\sigma(f(x^{T+1})-y^{T+1}) - \ell_\sigma(f^*(x^{T+1})-y^{T+1})d\rho_{\mathcal{Y}|\mathcal{X}}^{T+1}d\rho_{\mathcal{X}}^{T+1}\right|$$

$$+\left|\int_{\mathcal{X}}\int_{I_{Y^{T+1}}\cup II_{Y^{T+1}}}(f(x^{T+1})-y^{T+1})^2 - (f^*(x^{T+1})-y^{T+1})^2 d\rho_{\mathcal{Y}|\mathcal{X}}^{T+1}d\rho_{\mathcal{X}}^{T+1}\right|$$

$$\le \left|\int_{\mathcal{X}}\int_{I_{Y^{T+1}}}\ell_\sigma(f(x^{T+1})-y^{T+1}) - \ell_\sigma(f^*(x^{T+1})-y^{T+1})d\rho_{\mathcal{Y}|\mathcal{X}}^{T+1}d\rho_{\mathcal{X}}^{T+1}\right|$$

$$+\left|\int_{\mathcal{X}}\int_{I_{Y^{T+1}}}(f(x^{T+1})-y^{T+1})^2 - (f^*(x^{T+1})-y^{T+1})^2 d\rho_{\mathcal{Y}|\mathcal{X}}^{T+1}d\rho_{\mathcal{X}}^{T+1}\right|.$$

The last inequality is based on the fact that

$$|y^{T+1} - f(x^{T+1})| < |y^{T+1}| + \|f\|_\infty < \sigma, \ \forall (x^{T+1}, y^{T+1}) \in \mathcal{X} \times II_{Y^{T+1}}$$

and hence

$$\ell_\sigma(f(x^{T+1}) - y^{T+1}) = (f(x^{T+1}) - y^{T+1})^2.$$

Recalling that the Huber loss is $2\sigma$-Lipschitz continuous, then we get

$$\left| \int_\mathcal{X} \int_{I_{Y^{T+1}}} \ell_\sigma(f(x^{T+1}) - y^{T+1}) - \ell_\sigma(f^*(x^{T+1}) - y^{T+1}) d\rho_{\mathcal{Y}|\mathcal{X}}^{T+1} d\rho_\mathcal{X}^{T+1} \right|$$

$$\leq 2\sigma \left| \int_\mathcal{X} \int_{I_{Y^{T+1}}} |f(x) - f^*(x)| d\rho_{\mathcal{Y}|\mathcal{X}}^{T+1} d\rho_\mathcal{X}^{T+1} \right|$$

$$\leq 2\sigma \|f - f^*\|_\infty \text{Prob}\{I_Y\}.$$

According to Markov's inequality, for any constant $c > 0$, we further have

$$\text{Prob}\{I_{Y^{T+1}}\} = \text{Prob}\{|Y^{T+1}| \geq \sigma/2\} = \text{Prob}\{|Y^{T+1}|^{1+c} \geq (\sigma/2)^{1+c}\} \leq \frac{2^{1+c}\mathbb{E}(|Y^{T+1}|^{1+c})}{\sigma^{1+c}}.$$

Then we have

$$\left| \int_\mathcal{X} \int_{I_{Y^{T+1}}} \ell_\sigma(f(x^{T+1}) - y^{T+1}) - \ell_\sigma(f^*(x^{T+1}) - y^{T+1}) d\rho_{\mathcal{Y}|\mathcal{X}}^{T+1} d\rho_\mathcal{X}^{T+1} \right|$$

$$\leq \frac{2^{2+c}\|f - f^*\|_\infty \mathbb{E}(|Y^{T+1}|^{1+c})}{\sigma^c}.$$

Moreover, the second term in the right-hand side can be bounded by

$$\left| \int_\mathcal{X} \int_{I_{Y^{T+1}}} (f(x^{T+1}) - y^{T+1})^2 - (f^*(x^{T+1}) - y^{T+1})^2 d\rho_{\mathcal{Y}|\mathcal{X}}^{T+1} d\rho_\mathcal{X}^{T+1} \right|$$

$$\leq \|f - f^*\|_\infty \int_\mathcal{X} \int_{I_{Y^{T+1}}} |2y^{T+1} - f(x^{T+1}) - f^*(x^{T+1})| d\rho_{\mathcal{Y}|\mathcal{X}}^{T+1} d\rho_\mathcal{X}^{T+1}$$

$$\leq \|f - f^*\|_\infty \left[ \int_{I_{Y^{T+1}}} 2|y^{T+1}| d\rho_\mathcal{Y}^{T+1} + (\|f\|_\infty + \|f^*\|_\infty)\text{Prob}(I_{Y^{T+1}}) \right].$$

According to Holder inequality, we have

$$\int_{I_{Y^{T+1}}} |Y^{T+1}| d\rho_\mathcal{Y}^{T+1} \leq (\text{Prob}(I_{Y^{T+1}}))^{\frac{c}{1+c}} (\mathbb{E}|Y^{T+1}|^{1+c})^{\frac{1}{1+c}} \leq \frac{2^c \mathbb{E}(|Y^{T+1}|^{1+c})}{\sigma^c}.$$

Then we can deduce that

$$\|f - f^*\|_\infty \left[ \int_{I_{Y^{T+1}}} 2|y^{T+1}| d\rho_\mathcal{Y}^{T+1} + (\|f\|_\infty + \|f^*\|_\infty)\text{Prob}(I_{Y^{T+1}}) \right]$$

$$\leq \frac{(2^{2+c} + 2^{1+c})\|f - f^*\|_\infty \mathbb{E}(|Y^{T+1}|^{1+c})}{\sigma^c} + \frac{2^{1+c}(\|f\|_\infty + \|f^*\|_\infty)^2 \mathbb{E}(|Y^{T+1}|^{1+c})}{\sigma^{1+c}}$$

Finally, there holds

$$\left| \left[ \int_\mathcal{Z} \ell_\sigma(f(x^{T+1}) - Y^{T+1}) - \int_\mathcal{Z} \ell_\sigma(f^*(x^{T+1}) - Y^{T+1}) \right] - \|f - f^*\|_{L_2(\rho_\mathcal{X}^{T+1})}^2 \right| \lesssim \frac{M_c}{\sigma^c}$$

with

$$M_c = 2^{3+c}(M_{f^*} + M_{\hat{f}} + 1)^2 \mathbb{E}(|Y^{T+1}|^{1+c}).$$

This completes the proof. □

Define a stepping-stone function with respect to the distribution $\rho^{T+1}$:

$$f_{\eta,T+1}^* = \sum_{j=1}^p f_{\eta,T+1,j}^* = \arg \min_{f \in \mathcal{H}_K} \mathbb{E}\ell_\sigma(f(x^{T+1}) - Y^{T+1}) + \eta \sum_{j=1}^p \tau_j \|f_j\|_{K_j}^2.$$

To establish the bound of function approximation for stationary time series setting, we make the following error decomposition.

**Proposition 1.** *Let Assumptions 1-2 be true. For any $f \in \bar{\mathcal{H}}_Z$, there holds*

$$\|f - f^*\|^2_{L_2(\rho_{\mathcal{X}}^{T+1})} \leq E_1 + E_2 + E_3 + \frac{2M_c}{\sigma^c},$$

*where*

$$E_1 = \{\mathcal{E}_{T+1}^{\sigma}(f) - \mathcal{E}_T^{\sigma}(f)\} + \{\mathcal{E}_T^{\sigma}(f_{\eta,T+1}^*) - \mathcal{E}_{T+1}^{\sigma}(f_{\eta,T+1}^*)\}$$

$$E_2 = \mathcal{E}_{T+1}(f_{\eta,T+1}^*) - \mathcal{E}_{T+1}(f^*) + \eta \sum_{j=1}^{p} \tau_j \|f_{\eta,T+1,j}^*\|_{K_j}^2$$

*and*

$$E_3 = \{\mathcal{E}_T^{\sigma}(\hat{f}) + \lambda\Omega(\hat{f}) - \mathcal{E}_T^{\sigma}(\hat{f}_\eta) - \eta \sum_{j=1}^{p} \tau_j \|\hat{f}_{\eta,j}\|_{K_j}^2\}.$$

*Proof.* According to Lemma 2, for any $f \in \bar{\mathcal{H}}_Z$, we can make following error decomposition

$$\|f - f^*\|^2_{L_2(\rho_{\mathcal{X}}^{T+1})} = \mathcal{E}_{T+1}(f) - \mathcal{E}_{T+1}(f^*) \leq \mathcal{E}_{T+1}^{\sigma}(f) - \mathcal{E}_{T+1}^{\sigma}(f^*) + \frac{M_c}{\sigma^c}$$

$$= \underbrace{\{\mathcal{E}_{T+1}^{\sigma}(f) - \mathcal{E}_T^{\sigma}(f)\} + \{\mathcal{E}_T^{\sigma}(f_{\eta,T+1}^*) - \mathcal{E}_{T+1}^{\sigma}(f_{\eta,T+1}^*)\}}_{E_1} + \{\mathcal{E}_T^{\sigma}(f) - \mathcal{E}_T^{\sigma}(f_{\eta,T+1}^*)\}$$

$$+ \mathcal{E}_{T+1}^{\sigma}(f_{\eta,T+1}^*) - \mathcal{E}_{T+1}^{\sigma}(f^*) + \frac{M_c}{\sigma^c}$$

$$\leq E_1 + \underbrace{\mathcal{E}_{T+1}(f_{\eta,T+1}^*) - \mathcal{E}_{T+1}(f^*) + \eta \sum_{j=1}^{p} \tau_j \|f_{\eta,T+1,j}^*\|_{K_j}^2}_{E_2} + \{\mathcal{E}_T^{\sigma}(f) + \lambda\Omega(f) - \mathcal{E}_T^{\sigma}(f_{\eta,T+1}^*)$$

$$- \eta \sum_{j=1}^{p} \tau_j \|f_{\eta,T+1,j}^*\|_{K_j}^2\} + \frac{2M_c}{\sigma^c}$$

$$\leq E_1 + E_2 + \underbrace{\{\mathcal{E}_T^{\sigma}(f) + \lambda\Omega(f) - \mathcal{E}_T^{\sigma}(\hat{f}_\eta) - \eta \sum_{j=1}^{p} \tau_j \|\hat{f}_{\eta,j}\|_{K_j}^2\}}_{E_3} + \frac{2M_c}{\sigma^c}$$

$$\leq E_1 + E_2 + E_3 + \frac{2M_c}{\sigma^c}.$$

$\square$

In statistical machine learning community, we call $E_1$, $E_2$ and $E_3$ sample error, approximation error and hypothesis error, respectively. The sample error $E_1$ describes the divergence between the empirical risk $\mathcal{E}_T^{\sigma}(f)$ and the expected risk $\mathcal{E}^{\sigma}(f)$. The hypothesis error $E_2$ characterizes the difference between the empirical regularized risks with $\mathcal{H}_K$ and $\mathcal{H}_Z$. The approximation error measures the approximation ability of RKHS $\mathcal{H}_K$ to $\mathcal{H}$.

### B.1 SAMPLE ERROR

In this section, we focus on providing the bound of sample error $E_1$. Unfortunately, the traditional tools for complexity analysis such as covering number (Ron, 2000; Guo & Shi, 2011), Rademacher complexity (Mohri & Rostamizadeh, 2009)) and concentration inequalities ((Wu et al., 2007)), cannot be applied into this non-i.i.d. setting directly. To solve this problem, we employ the sequential Rademacher complexity developed in (Rakhlin et al., 2010; Kuznetsov & Mohri, 2020a). We define a function-based random variable as

$$\xi_f(z) = \ell_\sigma(f(x) - y) - \ell_\sigma(f_{\eta,T+1}^*(x) - y), f \in \bar{\mathcal{H}}_Z.$$

Now, we turn to establish the bound of

$$\mathbb{E}\xi_f(z^{T+1}) - \frac{1}{T} \sum_{t=1}^{T} \xi_f(z^t), \ \forall f \in \bar{\mathcal{H}}_Z.$$

A necessary ingredient needed for our analysis is data-dependent sequential Rademacher complexity Rakhlin et al. (2010), which we review in the following. We adopt the following definition of a complete binary tree: a $\mathcal{Z}$-valued complete binary tree $\mathbf{v}$ is a sequence $(v_1, ..., v_T)$ of $T$ mappings $v_t : \{\pm 1\}^{t-1} \to \mathcal{Z}, t \in [1, T]$. A path in the tree is $\gamma = (\gamma_1, ..., \gamma_{T-1}) \in \{\pm 1\}^{T-1}$. To simplify the notation, we will write $v_t(\gamma)$ instead $v_t(\gamma_1, ..., \gamma_{t-1})$, even though $v_t$ depends only on the first $t-1$ elements of $\gamma$. The following definition generalizes the classical notion of Rademacher complexity to sequential setting.

**Definition 3.** *The sequential Rademacher complexity $\mathcal{R}_T(G)$ of a function class $\mathcal{G}$ is defined by*

$$\mathcal{R}_T(\mathcal{G}) = \sup_{\mathbf{v}} \mathbb{E}[\sup_{g \in \mathcal{G}} \frac{1}{T} \sum_{t=1}^{T} \gamma_t g(v_t(\gamma))],$$

*where the supremum is taken over all complete binary trees of depth $T$ with values in $\mathcal{Z}$ and $\gamma$ is a sequence of Rademacher random variables.*

Based on the definition of sequential Rademacher complexity, we have the following concentration inequality:

**Lemma 3.** *Suppose that the time series $\{Z^t\}_{t=-\infty}^{\infty}$ is strictly stationary. For any $\delta > 0$, with probability at least $1 - \delta$, the following inequality holds for all $f \in \bar{\mathcal{H}}_Z$ and all $\alpha > 0$*

$$\mathbb{E}\xi_f(z^{T+1}) \leq \frac{1}{T} \sum_{t=1}^{T} \xi_f(z^t) + \frac{1}{\sqrt{T}} + 6M_{\ell_\sigma}\sqrt{4\pi \log T}\mathcal{R}_T(\mathcal{G}) + \frac{M_{\ell_\sigma}\sqrt{8\log \frac{1}{\delta}}}{\sqrt{T}},$$

*where $\mathcal{G} := \{\ell_\sigma(f, z) + \ell_\sigma(f_\eta, z) : f \in \bar{\mathcal{H}}_Z\}$ and $M_{\ell_\sigma}$ is a positive constant such that $\ell_\sigma(f(x^{(t)}) - y^{(t)}) \leq M_{\ell_\sigma}$ for any $f \in \bar{\mathcal{H}}_Z$ and $t \in \mathbb{Z}$.*

According to the property of sequential Rademacher complexity (see Proposition 14 in (Rakhlin et al., 2015)), we have

$$\mathcal{R}_T(\mathcal{G}) = \mathcal{R}_T(\ell_\sigma \circ \bar{\mathcal{H}}_Z, z) \leq 16\sigma(1 + 4\sqrt{2}\log^{3/2}(eT^2))\mathcal{R}_T(\bar{\mathcal{H}}_Z) \tag{11}$$

with the $2\sigma$-Lipschitz constant of $\ell_\sigma(\cdot)$.

**Lemma 4.** *Under Assumptions 1-2, there holds*

$$\mathcal{R}_T(\mathcal{G}) \leq \frac{32\sigma[1 + 4\sqrt{2}\log^{3/2}(eT^2)]M^2 p^{\frac{1}{2}}\kappa}{T}.$$

*Proof.* In fact, to every kernel $K_j$, we can associate a feature map $\phi$ with inner product $< \phi(x_j), \phi(x'_j) >= K(x_j, x'_j)$ for any $x_j, x'_j \in \mathcal{X}_j$ and $j = 1, ..., p$. Then the constrained data-dependent hypothesis space $\bar{\mathcal{H}}_Z$ can be rewritten as

$$\bar{\mathcal{H}}_Z = \{f(x) = \sum_{j=1}^{p}\sum_{t=1}^{T} \alpha_{ij}\phi(x_j^{(t)})\phi(x_j) : \sum_{j=1}^{p} \tau_j\|\alpha_j\|_2 \leq \frac{2M^2}{\lambda T \min_{j=1,..,p} \tau_j}\}.$$

By direct computation, we have

$$
\begin{aligned}
\mathcal{R}_T(\bar{\mathcal{H}}_Z) &= \sup_{\{\bar{x}_t,\bar{y}_t\}_{t=1}^T} \mathbb{E}_\sigma\Big[\sup_{\alpha\in\mathbb{R}^{Tp}} \frac{1}{T}\sum_{t=1}^T \sigma_t\big(\sum_{j=1}^p\sum_{i=1}^T \alpha_{ij}\phi(x_j^{(i)})\phi(\bar{x}_j^{(t)}(\sigma)) - y^{(t)}\big)\Big] \\
&= \frac{1}{T}\sup_{\{\bar{x}^{(t)}\}_{t=1}^T}\mathbb{E}_\sigma\sup_{\alpha\in\mathbb{R}^{Tp}}\sum_{t=1}^T\sigma_t\sum_{j=1}^p\sum_{i=1}^T\alpha_{ij}\phi(x_j^{(i)})\phi(\bar{x}_j^{(t)}(\sigma)) - \frac{1}{T}\sup_{\{\bar{y}^{(t)}\}_{t=1}^T}\mathbb{E}_\sigma\sum_{t=1}^T\sigma_t y^{(t)}(\sigma) \\
&= \frac{1}{T}\sup_{\{\bar{x}^{(t)}\}_{t=1}^T}\mathbb{E}_\sigma\sup_{\alpha\in\mathbb{R}^{Tp}}\sum_{t=1}^T\sigma_t\sum_{j=1}^p\sum_{i=1}^T\alpha_{ij}\phi(x_j^{(i)})\phi(\bar{x}_j^{(t)}(\sigma)) \\
&= \frac{1}{T}\sup_{\{\bar{x}^{(t)}\}_{t=1}^T}\mathbb{E}_\sigma\sup_{\alpha\in\mathbb{R}^{Tp}}\sum_{t=1}^T\sigma_t\sum_{j=1}^p\sum_{i=1}^T\alpha_{ij}K(x_j^{(i)},\bar{x}_j^{(t)}(\sigma)) \\
&= \frac{1}{T}\sup_{\{\bar{x}^{(t)}\}_{t=1}^T}\mathbb{E}_\sigma\sup_{\alpha\in\mathbb{R}^{Tp}}\alpha'\sum_{t=1}^T\sigma_t\mathbf{K}_t(\sigma),\ \ \mathbf{K}_t(\sigma)=(K(x_1^{(1)},\bar{x}_1^{(t)}(\sigma)),...,K(x_p^{(T)},\bar{x}_p^{(t)}(\sigma)))'\in\mathbb{R}^{Tp} \\
&\leq \frac{1}{T}\sup_{\alpha\in\mathbb{R}^{Tp}}\|\alpha'\|\sup_{\{\bar{x}_t\}_{t=1}^T}\mathbb{E}_\sigma\|\sum_{t=1}^T\sigma_t\mathbf{K}_t(\sigma)\|_2 \\
&\leq \frac{2M^2}{T^2}\sup_{\{\bar{x}^{(t)}\}_{t=1}^T}\mathbb{E}_\sigma\|\sum_{t=1}^T\sigma_t\mathbf{K}_t(\sigma)\|_2 \\
&= \frac{2M^2}{T^2}\sup_{\{\bar{x}^{(t)}\}_{t=1}^T}\sqrt{\mathbb{E}_\sigma\Big[\sum_{s,t=1}^T\sigma_t\sigma_s\mathbf{K}_t'(\sigma)\mathbf{K}_s(\sigma)\Big]} \\
&\leq \frac{2M^2}{T^2}\sup_{\{\bar{x}^{(t)}\}_{t=1}^T}\sqrt{\mathbb{E}_\sigma\Big[\sum_{t=1}^T\mathbf{K}_t'(\sigma)\mathbf{K}_t(\sigma)\Big]} \\
&\leq \frac{2M^2 p^{\frac{1}{2}}\kappa}{T}
\end{aligned}
$$

Thus, combining above result with Eq. (11). We get the desirable result. $\qquad\square$

Combining the results in Lemma 4 with the inequality in Lemma 3, we then obtain the bound of sample error $E_1$ for the strictly stationary time series setting.

**Proposition 2.** *Under Assumptions 1-2, for any $\delta > 0$, the following inequality holds for all $f \in \bar{\mathcal{H}}_Z$ and all $\alpha > 0$*

$$
E_1 \leq \frac{1}{\sqrt{T}} + 6M_{\ell_\sigma}\sqrt{4\pi\log T}R + \frac{M_{\ell_\sigma}\sqrt{8\log\frac{1}{\delta}}}{\sqrt{T}}
$$

*with probability at least $1 - \delta$, where*

$$
R = \frac{32\sigma[1 + 4\sqrt{2}\log^{3/2}(eT^2)]M^2 p^{\frac{1}{2}}\kappa}{T}.
$$

### B.2 APPROXIMATION ERROR

Since the output function $f_{\eta,T+1}^*$ is the optimal estimator in RKHS $\mathcal{H}_K$ for time $T + 1$, the learning rates of the learning algorithm indeed depend on the approximation ability of the hypothesis space $\mathcal{H}_K$ with respect to the optimal risk $\mathcal{E}(f^*)$ measured by the approximation error $E_2$. For any $j = 1, ..., p$ and $h \in \mathbb{Z}$, we first define the kernel integral operator $L_{K_j,T+1} : L_2(\rho_{\mathcal{X}_j}^{T+1}) \to L_2(\rho_{\mathcal{X}_j}^{T+1})$ associated with the kernel $K_j$ by

$$
L_{K_j,T+1}(f)(x_j^{T+1}) = \int_{\mathcal{X}_j} K_j(x_j^{T+1}, u_j)f(u_j)d\rho_{\mathcal{X}_j}^{T+1}(u_j), \forall h \in \mathbb{Z}.
$$

Note that $L_{K_j,T+1}$ is a compact and positive operator on $L_2(\rho_{\mathcal{X}_j}^{T+1})$. According to Mercer theorem, we can find the corresponding normalized eigenpairs $\{(\zeta_{h,i}^j, \psi_{h,i}^j)\}_{i\geq 1}$ such that $\{\psi_{h,i}^j\}_{i\geq 1}$ is an orthonormal basis of $L_2(\rho_{\mathcal{X}}^{T+1})$ and $\zeta_{h,i}^j \to 0$ as $i \to \infty$. Then for given $r > 0$, we defined the $r$-th power $L_{K_j,T+1}^r$ by

$$L_{K_j,T+1}^r(\sum_{i\geq 1} \beta_{h,i}^j \psi_{h,i}^j) = \sum_{i\geq 1} \beta_{h,i}^j (\zeta_{h,i}^j)^r \psi_{h,i}^j.$$

We introduce an intermediate function as follows:

$$\tilde{f}_{\eta,T+1,j} = (L_{K_j,T+1} + \eta\tau_j I)^{-1} L_{K_j,T+1} f_j^*, \forall j = 1, ..., p.$$

**Lemma 5.** *Under Assumption 3, for the intermediate function $\tilde{f}_{\eta,T+1,j}$ defined above, there holds*

$$\mathbb{E}\|\tilde{f}_{\eta,T+1,j}(x_j^{T+1}) - f_j^*(x_j^{T+1})\| + \eta\tau_j\|\tilde{f}_{\eta,T+1,j}(x_j^{T+1})\|_{K_j}^2 \leq 2(\eta\tau_j)^{2r}\|L_{K_j,T+1}^{-r} f_j^*\|_2^2.$$

*Proof.* Under Assumption 3, for any $h \in \mathbb{Z}$, we know that $f_j^* = L_{K_j,h}^r(g_{j,h}^*)$ for some $g_{h,j}^* = \sum_{i\geq 1} \beta_{h,i}\psi_{h,i}^j \in L_2(\rho_{\mathcal{X}_j}^{T+1})$. Then we have

$$f_j^* = L_{K_j,T+1}^r(\sum_{i\geq 1} \beta_{h,i}\psi_{h,i}^j) = \sum_{i\geq 1}(\zeta_{h,i}^j)^r \beta_{h,i}\psi_{h,i}^j.$$

The case $r = \frac{1}{2}$ means each $f_j^*$ lies in the RKHS $\mathcal{H}_{K_j}$. Then we have

$$\tilde{f}_{\eta,h,j} - f_j^* = (L_{K_j,h} + \eta\tau_j I)^{-1} L_{K_j,h} f_j^* - f_j^* = \sum_{i\geq 1} \frac{\eta\tau_j(\zeta_{h,i}^j)^r \beta_{h,i}^j \psi_{h,i}^j}{\zeta_i^j + \eta\tau_j}.$$

Then we have

$$\begin{aligned}
\|\tilde{f}_{\eta,T+1,j} - f_j^*\|_2^2 &= (\sum_{i\geq 1}\frac{\eta\tau_j(\zeta_{h,i}^j)^r\beta_{h,i}^j}{\zeta_{h,i}^j + \eta\tau_j})^2 = (\eta\tau_j)^{2r}\sum_{i\geq 1}(\frac{\eta\tau_j}{\zeta_{h,i}^j + \eta\tau_j})^{2-2r}(\frac{\zeta_{h,i}^j}{\zeta_{h,i}^j + \eta\tau_j})^{2r}(\beta_{h,i}^j)^2 \\
&\leq (\eta\tau_j)^{2r}\sum_{i\geq 1}(\beta_{h,i}^j)^2 = (\eta\tau_j)^{2r}\|L_{K_j,T+1}^{-r}f_j^*\|_2^2.
\end{aligned}$$

Similarly, we also have

$$\begin{aligned}
\eta\tau_j\|\tilde{f}_{\eta,T+1,j}\|_2^2 &= (\eta\tau_j)^2\sum_{i\geq 1}(\frac{(\zeta_{h,i}^j)^{1+r}}{\zeta_{h,i}^j + \eta\tau_j})^2(\beta_{h,i}^j)^2 = (\eta\tau_j)^{2r}\sum_{i\geq 1}(\frac{\zeta_{h,i}^j}{\zeta_{h,i}^j + \eta\tau_j})^{2+2r}(\frac{\eta\tau_j}{\zeta_{h,i}^j + \eta\tau_j})^{2-2r}(\beta_{h,i}^j)^2 \\
&\leq (\eta\tau_j)^{2r}\|L_{K_j,T+1}^{-r}f_j^*\|_2^2.
\end{aligned}$$

$\square$

**Proposition 3.** *Under Assumptions 3, there holds*

$$E_2 = \mathcal{E}(f_{\eta,T+1}^*) - \mathcal{E}(f^*) + \eta\sum_{j=1}^p \tau_j\|f_{\eta,T+1,j}^*\|_K^2 \leq (p+1)\sum_{j=1}^p(\eta\tau_j)^{2r}\|L_{K_j,T+1}^{-r}f_j^*\|_2^2.$$

*Proof.* According to Lemma 1, we have

$$\begin{aligned}
&\mathcal{E}_{T+1}(f_{\eta,T+1}^*) - \mathcal{E}_{T+1}(f^*) + \eta\sum_{j=1}^p \tau_j\|f_{\eta,T+1,j}^*\|_K^2 \\
\leq\ & \mathcal{E}_{T+1}^\sigma(f_{\eta,T+1}^*) - \mathcal{E}_{T+1}^\sigma(f^*) + \eta\sum_{j=1}^p \tau_j\|f_{\eta,T+1,j}^*\|_K^2 + \frac{M_c}{\sigma^c} \\
\leq\ & \mathcal{E}_{T+1}^\sigma(\tilde{f}_{\eta,T+1}) - \mathcal{E}_{T+1}^\sigma(f^*) + \eta\sum_{j=1}^p \tau_j\|\tilde{f}_{\eta,T+1,j}\|_K^2 + \frac{M_c}{\sigma^c} \\
\leq\ & \mathcal{E}_{T+1}(\tilde{f}_{\eta,T+1}) - \mathcal{E}_{T+1}(f^*) + \eta\sum_{j=1}^p \tau_j\|\tilde{f}_{\eta,T+1,j}\|_K^2 + \frac{2M_c}{\sigma^c}
\end{aligned}$$

Then we have

$$
\begin{aligned}
\mathcal{E}_{T+1}(\tilde{f}_{\eta,T+1}) - \mathcal{E}_{T+1}(f^*) + \eta \sum_{j=1}^{p} \tau_j \|\tilde{f}_{\eta,T+1,j}\|_K^2 &= \|\tilde{f}_{\eta,T+1} - f^*\|_2^2 + \eta \sum_{j=1}^{p} \tau_j \|\tilde{f}_{\eta,T+1,j}\|_K^2 \\
&\leq \|\sum_{j=1}^{p} [\tilde{f}_{\eta,h,j} - f_j^*]\|_2^2 + \eta \sum_{j=1}^{p} \tau_j \|\tilde{f}_{\eta,h,j}\|_K^2 \\
&\leq \sum_{j=1}^{p} p \|\tilde{f}_{\eta,T+1,j} - f_j^*\|_2^2 + \eta \sum_{j=1}^{p} \tau_j \|\tilde{f}_{\eta,T+1,j}\|_K^2 \\
&\leq (p+1) \sum_{j=1}^{p} (\eta\tau_j)^{2r} \|L_{K_j,T+1}^{-r} f_j^*\|_2^2.
\end{aligned}
$$

Combing above results, we get the desired result. $\qquad\square$

### B.3 Hypothesis Error

This section focuses on bounding the hypothesis error

$$
E_3 = \{\mathcal{E}_T^\sigma(\hat{f}) + \lambda\Omega(\hat{f}) - \mathcal{E}_T^\sigma(\hat{f}_\eta) - \eta \sum_{j=1}^{p} \|\hat{f}_{\eta,j}\|_K^2\}.
$$

We first give the properties of $\hat{f}$ and then use them to bridge $\hat{f}$ and $f_\eta$.

**Lemma 6.** *For all $j = 1, ..., p$, there holds*

$$
\tau_j \|\alpha_j^\eta\|_2 = \frac{1}{\eta T} \sqrt{\sum_{t=1}^{T} (\ell_\sigma'(y^t - \sum_{j=1}^{p} \boldsymbol{K}_{jt}' \alpha_j^\eta))^2} \leq \frac{2\sigma}{\eta T^{\frac{1}{2}}}
$$

*Proof.* Recall the represent theorem which ensures that

$$
\hat{f}_\eta = \sum_{j=1}^{p} \sum_{t=1}^{T} \alpha_{tj}^\eta K_j(x_j^t, \cdot), \ \alpha_{tj}^\eta \in \mathbb{R}.
$$

For notation simplicity, denote $\alpha_j^\eta = (\alpha_{1j}^\eta, ..., \alpha_{Tj}^\eta)' \in \mathbb{R}^T$ and $\boldsymbol{\alpha}^\eta = ((\alpha_1^\eta)', ..., (\alpha_p^\eta)')' \in \mathbb{R}^{Tp}$. From the definition of Eq. (7), we deduce that

$$
\boldsymbol{\alpha}^\eta = \arg\min_{\alpha \in \mathbb{R}^{Tp}} \{\frac{1}{T} \sum_{t=1}^{T} \ell_\sigma(y^t - \sum_{j=1}^{p} \boldsymbol{K}_{jt}' \alpha_j) + \eta \sum_{j=1}^{p} \tau_j (\alpha_j^\eta)' \boldsymbol{K}_j \alpha_j\},
$$

where $\boldsymbol{K}_j = \{K(x_j^s, x_j^i)\}_{i,s=1}^n \in \mathbb{R}^{T \times T}$. we then have

$$
\frac{1}{T} \sum_{t=1}^{T} \ell_\sigma'(y^t - \sum_{j=1}^{p} \boldsymbol{K}_{jt}' \alpha_j^\eta) \boldsymbol{K}_{ji}' = \eta\tau_j \boldsymbol{K}_j \alpha_j^\eta.
$$

It is easy to deduce that

$$
\alpha_j^\eta = \frac{1}{\eta\tau_j T} (\ell_\sigma'(y^1 - \sum_{j=1}^{p} \boldsymbol{K}_{j1}' \alpha_j^\eta), ..., \ell_\sigma'(y^T - \sum_{j=1}^{p} \boldsymbol{K}_{jT}' \alpha_j^\eta))'.
$$

Then we obtain that, for any $j = 1, ..., p$,

$$
\tau_j \|\alpha_j^\eta\|_2 = \frac{1}{\eta T} \sqrt{\sum_{t=1}^{T} (\ell_\sigma'(y^t - \sum_{j=1}^{p} \boldsymbol{K}_{jt}' \alpha_j^\eta))^2} \leq \frac{2\sigma}{\eta T^{\frac{1}{2}}}
$$

This completes the proof. $\qquad\square$

**Proposition 4.** *Under Assumptions 3, there holds*

$$E_3 \leq \lambda \Omega(\hat{f}_\eta) = \lambda \sum_{j=1}^{p} \tau_j \|\alpha_j^\eta\|_2 \leq \frac{2\lambda p \sigma}{\eta T^{\frac{1}{2}}}.$$

*Proof.*

$$
\begin{aligned}
E_3 &= \{\mathcal{E}_T^\sigma(\hat{f}) + \lambda \Omega(\hat{f}) - \mathcal{E}_T^\sigma(\hat{f}_\eta) - \eta \sum_{j=1}^{p} \|\hat{f}_{\eta,j}\|_K^2\} \\
&= \mathcal{E}_T^\sigma(\hat{f}) + \lambda \Omega(\hat{f}) - \mathcal{E}_T^\sigma(\hat{f}_\eta) - \lambda \Omega(\hat{f}_\eta) - \eta \sum_{j=1}^{p} \|\hat{f}_{\eta,j}\|_K^2 + \lambda \Omega(\hat{f}_\eta) \\
&\leq \lambda \Omega(\hat{f}_\eta)
\end{aligned}
$$

Combining the above inequality with Lemma 6, we get that

$$E_3 \leq \lambda \Omega(\hat{f}_\eta) = \lambda \sum_{j=1}^{p} \tau_j \|\alpha_j^\eta\|_2 \leq \frac{2\lambda p \sigma}{\eta T^{\frac{1}{2}}}.$$

$\square$

**Proof of Theorem 1:** Combining Propositions 1-4, for $T\eta \leq 1$, we have with confidence $1 - \delta$

$$\|f - f^*\|_{L_2(\rho_\mathcal{X}^{T+1})}^2 \leq \widetilde{C} \log(1/\delta)(T^{-\frac{1}{2}} + T^{-1}\sigma + \eta^{2r} + \lambda \eta^{-1} T^{-\frac{1}{2}}\sigma + \sigma^{-c}\lambda^{-2}T^{-2} + \sigma^{-c}\lambda^{-1}T^{-1}),$$

where $\widetilde{C}$ is a positive constant independently of $T, \lambda, \eta, \delta, \sigma$. By taking $\sigma = T^m, \eta = T^\beta$ and $\lambda = T^\gamma$, we then have with confidence $1 - \delta$

$$\|f - f^*\|_{L_2(\rho_\mathcal{X}^{T+1})}^2 \leq \widetilde{C} \log(1/\delta) T^{\Psi(m,\beta,\gamma,c,r)},$$

where

$$\Psi(m, \beta, \gamma, c, r) = \max\{-\frac{1}{2}, m - 1, 2r\beta, \gamma - \frac{1}{2} - \beta + m, -cm - 2\gamma - 2, -cm - \gamma - 1\}.$$

By taking $\sigma = \frac{1}{2c}, \beta = -\frac{1}{4r}$ and $\gamma = -\frac{1}{4r} - \frac{1}{2c}$ Direct computation shows that

$$\Psi(m, c, r) = \begin{cases} \max\{-\frac{1}{2}, \frac{1}{2c} - 1\}, & \text{if } m \leq 1 - \frac{1}{4r} \\ \max\{-\frac{1}{2}, m - 1, -cm + \frac{1}{2r} + 2m - 2\}, & \text{if } m > 1 - \frac{1}{4r}. \end{cases}$$

This completes the proof.

## C  PROOF OF THEOREM 2

Denote by $\mathbf{s} = \{s_t\}_{t=1}^T$ a probability set with $\sum_{t=1}^T s_t = 1$. We define following functions associated with the probability set $\mathbf{s}$:

$$f_{T+1}^* = \sum_{j=1}^{p} f_{T+1,j}^* = \arg \min_{f \in \mathcal{H}_K} \mathbb{E}\ell_\sigma(f(x^{T+1}) - y^{T+1}),$$

$$f_T^{*,\mathbf{s}} = \sum_{j=1}^{p} f_{T,j}^{*,\mathbf{s}} = \arg \min_{f \in \mathcal{H}_K} \sum_{t=1}^T s_t \ell_\sigma(f(x^t) - y^t),$$

$$\hat{f}_\eta^\mathbf{s} = \sum_{j=1}^{p} \hat{f}_{\eta,j}^\mathbf{s} = \arg \min_{f = \sum_{j=1}^p f_j, f_j \in \mathcal{H}_{K_j}} \{\sum_{t=1}^T s_t \ell_\sigma(y^t - \sum_{j=1}^p f_j(x_j^t)) + \eta \sum_{j=1}^p \tau_j \|f_j\|_{K_j}^2\},$$

and

$$\hat{f}^\mathbf{s} = \sum_{j=1}^{p} \hat{f}_j^\mathbf{s} = \arg \min_{f = \sum_{j=1}^p f_j, f_j \in \mathcal{H}_{K_j}} \{\sum_{t=1}^T s_t \ell_\sigma(y^t - \sum_{j=1}^p f_j(x_j^t)) + \lambda \Omega(f)\}.$$

Correspondingly, we denote by $\mathcal{E}_T^{\sigma,\mathbf{s}}(f) = \sum_{t=1}^T s_t \ell_\sigma(f(x^t) - y^t)$.

**Proposition 5.** *Let Assumptions 1-2 be true. For any $f \in \bar{\mathcal{H}}_Z$, there holds*

$$\|f - f^*\|^2_{L_2(\rho_{\mathcal{X}}^{T+1})} \le E_1 + E_2 + E_3 + \mathcal{E}_T^\sigma(f_T^*) + \eta \sum_{j=1}^p \tau_j \|f_{T,j}^*\|_{K_j}^2 + \frac{2M_c}{\sigma^c},$$

*where*

$$E_1 = \mathcal{E}_{T+1}^\sigma(\hat{f}^\mathbf{s}) - \mathcal{E}_T^{\sigma,\mathbf{s}}(\hat{f}^\mathbf{s})$$

$$E_2 = \mathcal{E}_T^{\sigma,\mathbf{s}}(\hat{f}^\mathbf{s}) + \lambda\Omega(\hat{f}^\mathbf{s}) - \mathcal{E}_T^{\sigma,\mathbf{s}}(\hat{f}_\eta^\mathbf{s}) - \eta \sum_{j=1}^p \tau_j \|\hat{f}_{\eta,j}^\mathbf{s}\|_{K_j}^2$$

*and*

$$E_3 = \mathcal{E}_{T+1}(f_{\eta,T+1}^*) - \mathcal{E}_{T+1}(f^*) + \eta \sum_{j=1}^p \tau_j \|f_{\eta,T+1,j}^*\|_{K_j}^2.$$

*Proof.* According to Lemma 2, for any $f \in \bar{\mathcal{H}}_Z$, we can make following error decomposition

$$
\begin{aligned}
&\|f - f^*\|^2_{L_2(\rho_{\mathcal{X}}^{T+1})} \\
=\ & \mathcal{E}_{T+1}(f) - \mathcal{E}_{T+1}(f^*) \\
\le\ & \mathcal{E}_{T+1}^\sigma(f) - \mathcal{E}_{T+1}^\sigma(f^*) + \frac{M_c}{\sigma^c} \\
\le\ & \underbrace{\{\mathcal{E}_{T+1}^\sigma(f) - \mathcal{E}_T^{\sigma,\mathbf{s}}(f)\}}_{E_1} + \{\mathcal{E}_T^{\sigma,\mathbf{s}}(f) - \mathcal{E}_T^{\sigma,\mathbf{s}}(f_T^{*,\mathbf{s}})\} + \mathcal{E}_T^{\sigma,\mathbf{s}}(f_T^{*,\mathbf{s}}) + \mathcal{E}_{T+1}^\sigma(f_{\eta,T+1}^*) - \mathcal{E}_{T+1}^\sigma(f^*) \\
& \qquad\qquad\qquad\qquad\qquad\qquad\qquad\qquad + \eta \sum_{j=1}^p \tau_j \|f_{\eta,T+1,j}^*\|_{K_j}^2\} + \frac{M_c}{\sigma^c} \\
\le\ & E_1 + \{\mathcal{E}_T^{\sigma,\mathbf{s}}(f) + \lambda\Omega(f) - \mathcal{E}_T^{\sigma,\mathbf{s}}(f_T^{*,\mathbf{s}}) - \eta \sum_{j=1}^p \tau_j \|f_{T,j}^{*,\mathbf{s}}\|_{K_j}^2\} + \mathcal{E}_T^{\sigma,\mathbf{s}}(f_T^{*,\mathbf{s}}) + \eta \sum_{j=1}^p \tau_j \|f_{T,j}^{*,\mathbf{s}}\|_{K_j}^2 \\
& \qquad\qquad\qquad + \mathcal{E}_{T+1}^\sigma(f_{\eta,T+1}^*) - \mathcal{E}_{T+1}^\sigma(f^*) + \eta \sum_{j=1}^p \tau_j \|f_{\eta,T+1,j}^*\|_{K_j}^2\} + \frac{M_c}{\sigma^c} \\
\le\ & E_1 + \underbrace{\{\mathcal{E}_T^{\sigma,\mathbf{s}}(f) + \lambda\Omega(f) - \mathcal{E}_T^{\sigma,\mathbf{s}}(\hat{f}_\eta^\mathbf{s}) - \eta \sum_{j=1}^p \tau_j \|\hat{f}_{\eta,j}^\mathbf{s}\|_{K_j}^2\}}_{E_2} + \mathcal{E}_T^{\sigma,\mathbf{s}}(f_T^{*,\mathbf{s}}) + \eta \sum_{j=1}^p \tau_j \|f_{T,j}^{*,\mathbf{s}}\|_{K_j}^2 \\
& \qquad\qquad\qquad + \mathcal{E}_{T+1}^\sigma(f_{\eta,T+1}^*) - \mathcal{E}_{T+1}^\sigma(f^*) + \eta \sum_{j=1}^p \tau_j \|f_{\eta,T+1,j}^*\|_{K_j}^2\} + \frac{M_c}{\sigma^c} \\
\le\ & E_1 + E_2 + \underbrace{\mathcal{E}_{T+1}^\sigma(f_{\eta,T+1}^*) - \mathcal{E}_{T+1}^\sigma(f^*) + \eta \sum_{j=1}^p \tau_j \|f_{\eta,T+1,j}^*\|_{K_j}^2}_{E_3} + \mathcal{E}_T^{\sigma,\mathbf{s}}(f_T^{*,\mathbf{s}}) + \eta \sum_{j=1}^p \tau_j \|f_{T,j}^{*,\mathbf{s}}\|_{K_j}^2 + \frac{2M_c}{\sigma^c} \\
\le\ & E_1 + E_2 + E_3 + \mathcal{E}_T^{\sigma,\mathbf{s}}(f_T^{*,\mathbf{s}}) + \eta \sum_{j=1}^p \tau_j \|f_{T,j}^{*,\mathbf{s}}\|_{K_j}^2 + \frac{2M_c}{\sigma^c} \\
\le\ & E_1 + E_2 + E_3 + \min_{f \in \mathcal{H}_Z} \{\mathcal{E}_T^{\sigma,\mathbf{s}}(f) + \lambda\Omega(f)\} + \eta \sum_{j=1}^p \tau_j \|f_{T,j}^{*,\mathbf{s}}\|_{K_j}^2 + \frac{2M_c}{\sigma^c}.
\end{aligned}
$$

$\square$

Following the generalization analysis in (Kuznetsov & Mohri, 2020a), we need to discrepancy measure to measure the discrepancy of the non-stationarity of the stochastic process $\{Z_t\}_{t=-\infty}^\infty$ with respect to both the loss function $\ell_\sigma$ and the hypothesis set $\bar{\mathcal{H}}_Z$.

**Definition 4.** *The discrepancy describes the discrepancy between target distribution and the distribution of the sample. For any $f \in \mathcal{H}_Z$, the discrepancy measure with respect to Huber loss is defined as*

$$\text{disc}(\mathbf{s}) := \sup_{f \in \mathcal{H}_Z} \left\{ \mathbb{E}\ell_\sigma(f(x^{T+1}) - y^{T+1}) - \sum_{t=1}^{T} \mathbb{E}s_t\ell_\sigma(f(x^t) - y^t) \right\}.$$

**Lemma 7.** *For any $\delta > 0$, with probability at least $1 - \delta$, the following inequality holds for all $f \in \bar{\mathcal{H}}_Z$ and all $\alpha > 0$*

$$\mathbb{E}\ell_\sigma(f(x^{T+1}) - y^{T+1}) \leq \sum_{t=1}^{T} s_t\ell_\sigma(f(x^t) - y^t) + \text{disc}(\mathbf{s}) + \|\mathbf{s}\|_2 + 6M_{\ell_\sigma}\sqrt{4\pi \log T}\mathcal{R}_T(\mathcal{G}) + M_{\ell_\sigma}\|\mathbf{s}\|_2\sqrt{8\log\frac{1}{\delta}},$$

*where $\mathcal{G} := \{\ell_\sigma(f, z) : f \in \bar{\mathcal{H}}_Z\}$ and $M_{\ell_\sigma}$ is a positive constant such that $\ell_\sigma(f(x^t) - y^t) \leq M_{\ell_\sigma}$ for any $f \in \mathcal{H}_Z$ and $t \in \mathbb{Z}$.*

**Proof of Theorem 2** The proofs of bounding errors $E_1$, $E_2$ and $E_3$ proceeds similarly to the proof of Theorem 1 and are omitted for brevity.

According to Lemma 3, we have

$$E_1 \leq \text{disc}(\mathbf{s}) + \|\mathbf{s}\|_2 + 6M_{\ell_\sigma}\sqrt{4\pi \log T}\mathcal{R} + M_{\ell_\sigma}\|\mathbf{s}\|_2\sqrt{8\log\frac{1}{\delta}},$$

where

$$R = \|\mathbf{s}\|_2[32\sigma[1 + 4\sqrt{2}\log^{3/2}(eT^2)]M^2p^{\frac{1}{2}}\kappa].$$

Moreover, from Proposition 4, we can get

$$E_3 \leq (p+1)\sum_{j=1}^{p}(\eta\tau_j)^{2r}\|L_{K_j,T+1}^{-r}f_j^*\|_2^2.$$

Similarly, to bound the hypothesis error

$$E_2 = \{\mathcal{E}_T^{\sigma,\mathbf{s}}(\hat{f}^{\mathbf{s}}) + \lambda\Omega(\hat{f}^{\mathbf{s}}) - \mathcal{E}_T^{\sigma,\mathbf{s}}(\hat{f}_\eta^{\mathbf{s}}) - \eta\sum_{j=1}^{p}\|\hat{f}_{\eta,j}^{\mathbf{s}}\|_K^2\},$$

we first give the properties of $\hat{f}$ and then use them to bridge $\hat{f}$ and $f_\eta$.

**Lemma 8.** *For all $j = 1, ..., p$, there holds*

$$E_2 \leq \frac{2\lambda p\sigma\|\mathbf{s}\|_2}{\eta}.$$

*Proof.* Recall the represent theorem which ensures that

$$\hat{f}_\eta^{\mathbf{s}} = \sum_{j=1}^{p}\sum_{t=1}^{T}\alpha_{tj}^{\eta,\mathbf{s}}K_j(x_j^t, \cdot), \ \alpha_{tj}^{\eta,\mathbf{s}} \in \mathbb{R}.$$

For notation simplicity, denote $\alpha_j^{\eta,\mathbf{s}} = (\alpha_{1j}^{\eta,\mathbf{s}}, ..., \alpha_{Tj}^{\eta,\mathbf{s}})' \in \mathbb{R}^T$ and $\boldsymbol{\alpha}^{\eta,\mathbf{s}} = ((\alpha_1^{\eta,\mathbf{s}})', ..., (\alpha_p^{\eta,\mathbf{s}})')' \in \mathbb{R}^{Tp}$. We then deduce that

$$\boldsymbol{\alpha}^{\eta,\mathbf{s}} = \underset{\alpha \in \mathbb{R}^{Tp}}{\arg\min}\{\sum_{t=1}^{T}s_t\ell_\sigma(y^t - \sum_{j=1}^{p}\boldsymbol{K}_{jt}'\alpha_j) + \eta\sum_{j=1}^{p}\tau_j(\alpha_j^\eta)'\boldsymbol{K}_j\alpha_j\},$$

where $\boldsymbol{K}_j = \{K(x_j^s, x_j^i)\}_{i,s=1}^{n} \in \mathbb{R}^{T \times T}$. We get

$$\sum_{t=1}^{T}s_t\ell_\sigma'(y^t - \sum_{j=1}^{p}\boldsymbol{K}_{jt}'\alpha_j^{\eta,\mathbf{s}})\boldsymbol{K}_{ji}' = \eta\tau_j\boldsymbol{K}_j\alpha_j^{\eta,\mathbf{s}}.$$

It is easy to deduce that

$$\alpha_j^{\eta,\mathbf{s}} = \frac{1}{\eta\tau_j}(s_1\ell_\sigma'(y^1 - \sum_{j=1}^p \mathbf{K}_{j1}'\alpha_j^\eta), ..., s_T\ell_\sigma'(y^T - \sum_{j=1}^p \mathbf{K}_{jT}'\alpha_j^\eta))'.$$

Then we obtain that, for any $j = 1, ..., p$,

$$\tau_j\|\alpha_j^\eta\|_2 = \frac{1}{\eta}\sqrt{\sum_{t=1}^T s_t^2(\ell_\sigma'(y^t - \sum_{j=1}^p \mathbf{K}_{jt}'\alpha_j^\eta))^2} \leq \frac{2\sigma\|\mathbf{s}\|_2}{\eta}$$

This completes the proof. $\qquad\qquad\square$

Under Assumptions 1-2, there holds

$$
\begin{aligned}
E_2 &= \{\mathcal{E}_T^\sigma(f) + \lambda\Omega(\hat{f}^{\mathbf{s}}) - \mathcal{E}_T^\sigma(\hat{f}_\eta^{\mathbf{s}}) - \eta\sum_{j=1}^p \|\hat{f}_{\eta,j}^{\mathbf{s}}\|_K^2\} \\
&= \mathcal{E}_T^\sigma(\hat{f}^{\mathbf{s}}) + \lambda\Omega(\hat{f}^{\mathbf{s}}) - \mathcal{E}_T^\sigma(\hat{f}_\eta^{\mathbf{s}}) - \lambda\Omega(\hat{f}_\eta^{\mathbf{s}}) - \eta\sum_{j=1}^p \|\hat{f}_{\eta,j}^{\mathbf{s}}\|_K^2 + \lambda\Omega(\hat{f}_\eta^{\mathbf{s}}) \\
&\leq \lambda\Omega(\hat{f}_\eta^{\mathbf{s}}) = \lambda\sum_{j=1}^p \tau_j\|\alpha_j^{\eta,\mathbf{s}}\|_2 \\
&\leq \frac{2\lambda p\sigma\|\mathbf{s}\|_2}{\eta}.
\end{aligned}
$$

By combining the above results, we have with confidence $1 - \delta$

$$
\begin{aligned}
&\|\hat{f}^{\mathbf{s}} - f^*\|_{L_2(\rho_\mathcal{X}^{T+1})}^2 \\
\leq~ &\mathrm{disc}(\mathbf{s}) + \min_{f\in\mathcal{H}_Z}\{\mathcal{E}_T^{\sigma,\mathbf{s}}(f) + \lambda\Omega(f)\} \\
&+ \widetilde{C}\log(1/\delta)(\|\mathbf{s}\|_2 + \|\mathbf{s}\|_2\sigma + \eta^{2r} + \lambda\eta^{-1}\|\mathbf{s}\|_2\sigma + \sigma^{-c}\lambda^{-2}T^{-2} + \sigma^{-c}\lambda^{-1}T^{-1} + \eta),
\end{aligned}
$$

where $\widetilde{C}$ is a positive constant independently of $T, \lambda, \eta, \delta, \sigma$ and $p$. By taking $\sigma = T^{\frac{1}{2c}}$, $\lambda = T^{-1}$ and $\eta = T^{-\frac{1}{2}}$, we completes the proof.

## D  PROOF OF THEOREM 3

According to the definition of $\mathrm{disc}(T + 1)$, we have

$$
\begin{aligned}
\mathrm{disc}(\mathbf{s}) &= \sup_{f\in\mathcal{H}_Z}\left\{\mathbb{E}\ell_\sigma(f(x^{T+1}) - y^{T+1}) - \sum_{t=1}^T s_t\mathbb{E}\ell_\sigma(f(x^t) - y^t)\right\} \\
&\leq \sup_{f\in\mathcal{H}_Z}[\sum_{t=1}^T q_t^*\mathbb{E}\ell_\sigma(f(x^t) - y^t) - \sum_{t=1}^T s_t\mathbb{E}\ell_\sigma(f(x^t) - y^t)] + \mathrm{disc}(\mathbf{q}^*).
\end{aligned}
$$

Furthermore, it is easy to deduce that

$$
\begin{aligned}
&\sup_{f\in\mathcal{H}_Z}\sum_{t=1}^T (q_t^* - s_t)\mathbb{E}\ell_\sigma(f(x^t) - y^t) - \sup_{f\in\mathcal{H}_Z}\sum_{t=1}^T (q_t^* - s_t)\ell_\sigma(f(x^t) - y^t) \\
\leq~ &\sup_{f\in\mathcal{H}_Z}\sum_{t=1}^T (q_t^* - s_t)[\mathbb{E}\ell_\sigma(f(x^t) - y^t) - \ell_\sigma(f(x^t) - y^t)].
\end{aligned}
$$

**Proof of Theorem 3** According to the proof of Theorem 1 in Kuznetsov & Mohri (2020a), we have

$$\sup_{f\in\mathcal{H}_Z}\sum_{t=1}^T (q_t^* - s_t)[\mathbb{E}\ell_\sigma(f(x^t) - y^t) - \ell_\sigma(f(x^t) - y^t)] \leq \frac{1}{\sqrt{T}} + 6M_{\ell_\sigma}\sqrt{4\pi\log T}\mathcal{R}_T(\mathcal{G}) + M_{\ell_\sigma}\|\mathbf{q}^* - \mathbf{s}\|_2\sqrt{8\log\frac{1}{\delta}}$$

By combining above results, we have with confidence $1 - \delta$

$$\|\hat{f}^{\mathbf{s}} - f^*\|_{L_2(\rho_{\mathcal{X}}^{T+1})}^2$$

$$\leq \quad \operatorname{disc}(\mathbf{q}^*) + \sup_{f \in \mathcal{H}_Z} \sum_{t=1}^{T} (q_t^* - s_t)\ell_\sigma(f(x^t) - y^t) + \min_{f \in \mathcal{H}_Z}\{\mathcal{E}_T^{\sigma,\mathbf{s}}(f) + \lambda\Omega(f)\}$$

$$+ \widetilde{C}\log(1/\delta)(\|\mathbf{q}^* - \mathbf{s}\|_2 + \|\mathbf{s}\|_2 + \|\mathbf{s}\|_2\sigma + \eta^{2r} + \lambda\eta^{-1}\|\mathbf{s}\|_2\sigma + \sigma^{-c}\lambda^{-2}T^{-2} + \sigma^{-c}\lambda^{-1}T^{-1} + \eta).$$

By taking $\sigma = T^{\frac{1}{2c}}$, $\lambda = T^{-1}$ and $\eta = T^{-\frac{1}{2}}$, we complete the proof.

---

**Algorithm 1:** Optimization procedure for adaptive SpHAM

---

**Input**: Data $\{(x^t, y^t)\}_{t=1}^T$, Max-Iter $Z \in \mathbb{Z}$, Mercer kernel $K_j, j = 1, ..., p$ with bandwidth $d$,
  Weights $\tau_l, l = 1, ..., p$, $\mathbf{q}^*$.
**Initialization**: Lipschitz constant $L$, $\mathbf{s}^0$.
**Step A: Computing weights $\hat{\mathbf{s}}$:**
**for** $z = 1, ..., Z$ **do**
  | 1. Compute $A^{z-1}$ via DC-programming (or gradient descent method);
  | 2. Update $\mathbf{s}^z$ via (12).
**Output:** $\hat{\mathbf{s}} = \mathbf{s}^Z$.
**Step B: Computing $\hat{f}^{\mathbf{s}}$:**
**for** $z = 1, ..., Z$ **do**
  | 1): Compute $\alpha^z = p_L(\beta^z)$ via (14);
  | 2): $m_{z+1} = \frac{1+\sqrt{1+4m_z^2}}{2}$;
  | 3): $\beta^{z+1} = \alpha^z + \frac{m_z-1}{m_{z+1}}(\alpha^z - \alpha^{z-1})$.
**Output:** $\alpha^{\hat{\mathbf{s}}} = \alpha^Z$;
**Prediction function:** $\hat{f}^{\mathbf{s}} = \sum_{j=1}^P \sum_{t=1}^T \alpha_{tj}^{\hat{\mathbf{s}}} K_j(x_j^t, \cdot)$;
**Variable selection:** $\{j : \|\alpha_j^{\mathbf{s}}\|_2 \geq v, j = 1, ..., p\}$.

---

## E  EXPERIMENT OPTIMIZATION

The optimization problem (10) reduces to problem 7 when taking $\mathbf{s}^* = \frac{1}{T}\mathbb{I}_T$. Recall the optimization problem in **Step A**:

$$\hat{\mathbf{s}} = \arg\min_{\mathbf{s}}\{ \sup_{f \in \mathcal{H}_Z} \sum_{t=1}^{T} (q_t^* - s_t)\ell_\sigma(f(x^t) - y^t) + \frac{\lambda_1}{2}\|\mathbf{q}^* - \mathbf{s}\|_2^2 + \frac{\lambda_2}{2}\|\mathbf{s}\|_2^2 T^{\frac{1}{2c}} \}.$$

The above optimization problem can be equivalently rewritten as a common type of bilevel optimization problem (Colson et al., 2007), i.e,

Outer problem: $\min_{\mathbf{s}} \sum_{t=1}^T (q_t^* - s_t)\ell_\sigma(f_{\mathbf{s}}(x_t) - y_t) + \lambda_1\|\mathbf{q}^* - \mathbf{s}\|_2^2 + \lambda_2 T^{1/2c}\|\mathbf{s}\|_2^2$,

Inner problem: $f_{\mathbf{s}} = \arg\max_{f \in \mathcal{H}_{\mathbf{z}}} \sum_{t=1}^T (q_t^* - s_t)\ell_\sigma(f(x_t) - y_t)$,

where the outer (min) problem parameterized by $\mathbf{s}$, is nested within the inner (max) problem. The outer problem can be solved by standard gradient descent, where in each step, we need to optimize the inner (max) problem with last updated $\mathbf{s}$.

We denote by $k$ the iteration time. Then for $k + 1$-th iteration, we have the following gradient update rule

$$\mathbf{s}^{k+1} = s^k - \gamma(A^k - \lambda_1(\mathbf{q}^* - \mathbf{s}^k) + \lambda_2 T^{\frac{1}{2c}}\mathbf{s}^k), \qquad (12)$$

where $\gamma$ is learning rate,

$$A^k = -\left(\ell_\sigma(\sum_{t=1}^T \sum_{j=1}^p \alpha_{tj}^{\mathbf{s}^k} K_j(x_j^t, x_j^1) - y^1), ..., \ell_\sigma(\sum_{t=1}^T \sum_{j=1}^p \alpha_{tj}^{\mathbf{s}^k} K_j(x_j^t, x_j^T) - y^T)\right)' \in \mathbb{R}^T$$

and

$$\alpha^{\mathbf{s}^k} = (\alpha_{11}^{\mathbf{s}^k}, ..., \alpha_{T1}^{\mathbf{s}^k}, ..., \alpha_{1p}^{\mathbf{s}^k}, ..., \alpha_{Tp}^{\mathbf{s}^k})' \in \mathbb{R}^{Tp}$$

is obtained by the following weighted optimization problem

$$\alpha^{\mathbf{s}^k} = \arg\max_{\alpha} \sum_{t=1}^{T} (q_t^* - s_t^k)\ell_\sigma(\sum_{m=1}^{T}\sum_{j=1}^{p} \alpha_{mj} K_j(x_j^m, x_j^t) - y^t).$$

Note that the inner problem is subjected to $f \in \mathcal{H}_K$, i.e., $\|\alpha\|_2 = \sum_{j=1}^{p} \tau_j \|\alpha_j\|_2 \leq \frac{2M^2}{\lambda T \min_{j=1,...,p} \tau_j}$. A widely-used method for solving this constrained inner problem is DC programming (Tao & An, 1998). For simplicity, we here transform this inner problem into a regularized problem with $\ell_2$-norm regularizer, and solve this regularized problem by standard gradient method. After obtaining the solution $\hat{\mathbf{s}}$, we turn to solve the following weighted optimization problem in **Step B**:

$$\alpha^{\hat{\mathbf{s}}} = \arg\min_{\alpha_j \in \mathbb{R}^T, j=1,...,p} \{\sum_{t=1}^{T} \hat{s}_t \ell_\sigma(y^t - \sum_{j=1}^{p} (\mathbf{K}_j^t)'\alpha_j) + \lambda \sum_{j=1}^{p} \tau_j \|\alpha_j\|_2\}. \tag{13}$$

We can see that this problem contains non-smooth function $\lambda \sum_{j=1}^{p} \tau_j \|\alpha_j\|_2$ which makes the standard gradient descent method inapplicable. To conquer this challenge, we leverage fast iterative shrinkage-thresholding algorithm (FISTA)(Beck & Teboulle, 2009). Our optimization problem becomes

$$\alpha^k = p_L(\alpha^{k-1}) := \arg\min_{\alpha} \left\{ \lambda \sum_{j}^{p} \tau_j \|\alpha_j\|_2 + \frac{L}{2}\|\alpha - (\alpha^{k-1} - \frac{1}{L}\sum_{t=1}^{T} \hat{s}_t \nabla\ell_\sigma(y^t - (K^t)'\alpha^{k-1}))\|^2 \right\}$$

$$= \left(\left(1 - \frac{\lambda}{\|v_j\|_2}\right)_+ v_j\right)_{1\leq j\leq p}, \tag{14}$$

where

$$v_j = \alpha_j^{k-1} - \frac{1}{L}\sum_{t=1}^{T} \hat{s}_t \nabla_{\alpha_j}\ell_\sigma(y^t - (K^t)'\alpha^{k-1})$$

and

$$\nabla_{\alpha_j}\ell_\sigma(y^t - (K^t)'\alpha^{k-1}) = \begin{cases} 2((K^t)'\alpha - y^t)(K_j^t)' & |(K^t)'\alpha - y^t| < \sigma \\ 2\sigma & (K^t)'\alpha - y^t \geq \sigma \\ -2\sigma & (K^t)'\alpha - y^t \leq -\sigma \end{cases}$$

and $L = \max \frac{1}{T}\sum_{t=1}^{T} \|(K^t)'K^t\|$ is the lipschitz constant $\frac{1}{T}\sum_{t=1}^{T} \nabla\ell_\sigma(y^t - (K^t)'\alpha^{k-1})$. Denote Proj($\mathbf{s}$) as a projection of $\mathbf{s}$ and $v > 0$ as the threshold value. To focus on the weights of samples which are really useful for forecasting, we consider Proj($\mathbf{s}$) as a box projection such that $s_t > q_t^*, \forall t = 1, ..., p$). Finally, the optimization procedure for adaptive SpHAM can be summarized in Algorithm 1. Note that if we only run **Step B** with $\hat{s}_t = \frac{1}{T}, \forall t = 1, ..., T$, we can further obtain the optimization procedure for SpHAM.

**Remark 3.** *The computational complexity of Algorithm 1 depends on the optimization strategy for DC programming, the training size $T$, dimension $p$ and the iteration times $Z$. We denote by the computational complexity of DC programming $O(DC(T, p))$. Then, the computational complexity of Step A is $O(ZDC(T, p) + ZT)$ and the computational complexity of Step B is $O(ZT^3p^3)$. Thus, the total computational complexity of Algorithm 1 is $O(ZO(DC) + ZT^3p^3)$. For large scale data, we can further speed up Algorithm 1 by random Fourier features technique [Rahimi and Recht 2007], which is leaved for future work. This has been carefully discussed in the revised manuscript.*

# F ADDITIONAL EXPERIMENT

## F.1 EVALUATION ON BENCHMARK DATA

We test our algorithm on nonlinear dataset from CauseMe. The hyper-parameter selection is the same as the one in Section 4. The results in Table 3 verify the effectiveness of our method.

Table 3: The ASE on CauseMe data ($p$ refers to the dimension of features).

| Methods | $(p = 3, T = 300)$ | $(p = 5, T = 300)$ |
|---------|---------|---------|
| LSTM | 0.7681 | **0.9230** |
| TS$-$SpAM | 0.7782 | 0.9485 |
| SpHAM | **0.7548** | 0.9484 |

### F.2 EXPERIMENTS ON AIR QUALITY DATASET

We use the Air Quality dataset obtained from UCI Machine Learning Repository (`https://archive.ics.uci.edu/ml/datasets/Air+quality`) to test our model's ability to detect the Granger causality. This dataset includes 9358 hourly air quality data in an Italian city, collected from March 2004 to February 2005. The details of the dataset can be obtained on the UCI website. The Granger causal network we detect is shown in Figure 3. Given the network, we observe that the temperature (T) influences ozone ($O_3$) and nitrogen dioxide ($NO_2$) which is validated in Kalisa et al. (2018). Mwaniki et al. (2014) confirm the relationship between relative humidity (RH) and nitrogen dioxide ($NO_2$). Yan et al. (2018) verify the relationship between humidity and nitrogen dioxide ($NO_2$).

### F.3 EXPERIMENTS ON CORONAL MASS EJECTIONS DATASET

Coronal Mass Ejections (CMEs) are the most violent eruptions in the Solar System. Despite machine learning approaches have been applied to these tasks recently Wang et al. (2019); Liu et al. (2018), there is no any work for interpretable prediction with Granger causal network. CMEs data are provided in The Richardson and Cane List (`http://www.srl.caltech.edu/ACE/ASC/DATA/level3/icmetable2.htm`). From this link, we collect 152 ICMEs observations from 1996 to 2016. The features of CMEs are provided in SOHO LASCO CME Catalog (`https://cdaw.gsfc.nasa.gov/CME_list/`). In-situ solar wind parameters can be downloaded from OMNIWeb Plus (`https://omniweb.gsfc.nasa.gov/`). A total of 9 features are chosen as input, including: (1) Central PA (CPA), (2) Angular Width, (3) three approximated speeds ( Linear Speed, 2nd-order Speed at final height,and 2nd-order Speed at 20 Rs), (4) Mass, (5) Kinetic Energy, (6) MPA and (7)CMEs arrival time. Figure 4 shows that Granger causal network when the output is CMEs Arrival time. Some interesting findings are concluded from this Granger causal network. For instance, Speed and Mass, as the significant variables causing CMEs arrival time, have been also screened out in Liu et al. (2018). Morover, the CMEs Angular Width does not cause the CMEs arrival time forecasting, while Liu et al. (2018) state that they have a significant correlation. This indicates that the CMEs arrival time is affected by the Angular Width at current time, but not by the historical Angular Width.

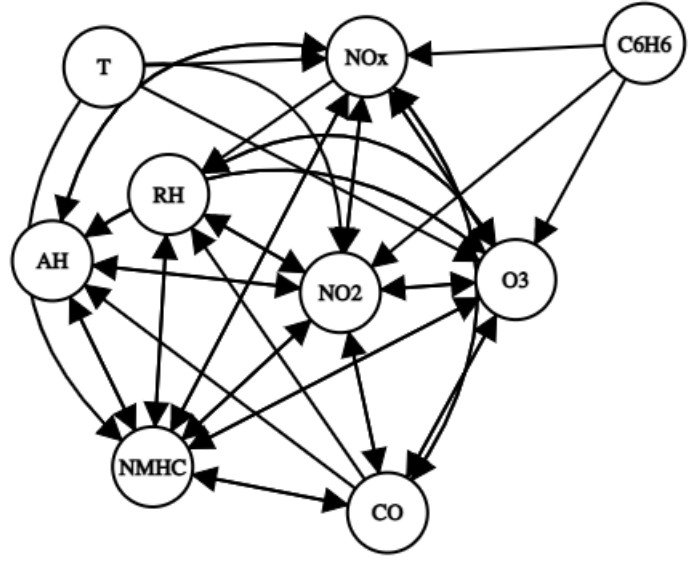

Figure 3: The Granger causal network on Air Quality dataset.

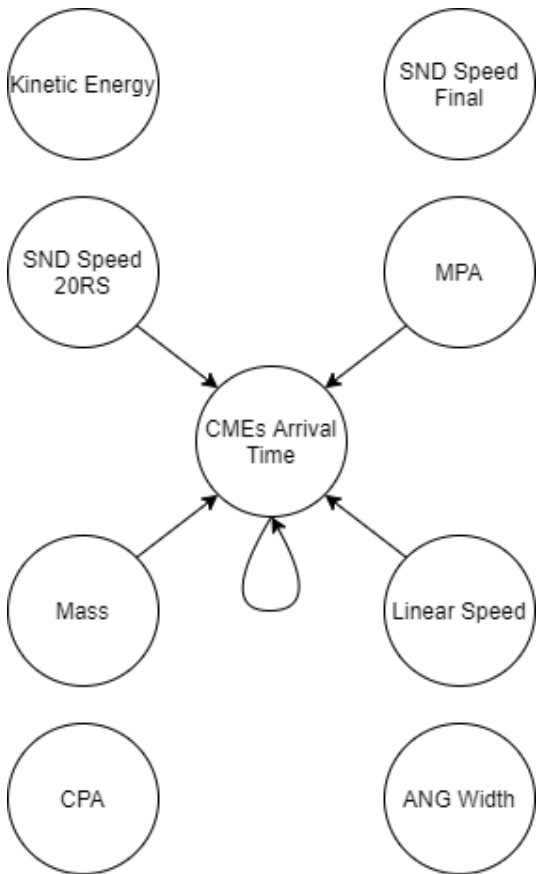

Figure 4: The Granger causal network on CMEs dataset.

