# OpenReview forum: "Huber Additive Models for Non-stationary Time Series Analysis"
_ICLR.cc/2022/Conference — ICLR 2022 Poster_

### Official Review · Reviewer_4Rek · 2021-10-31

**Correctness:** 3
**Technical Novelty And Significance:** 3
**Empirical Novelty And Significance:** 2
**Recommendation:** 6
**Confidence:** 4

**Main Review:**

Strengths:
1. The proposed time series model is both robust and interpretable.
2. Some novel performance lower bounds for the proposed model.

Weaknesses:
1. Although the simple linear additive model is well interpretable, its modeling capacity is quite weak on the other hand. As a remedy not destroying your optimization framework, you may consider adding some interacting terms, like Y2*Y3.
2. The proposed robust time series prediction model with the three theorems constitutes the main contribution of the paper. But to me, the meanings of the lower bound (namely each part of the right-hand side) are hard to interpret. Are they tight bounds in general? Besides, are $\hat{f}$ solutions of the corresponding optimization problems?
3. It is well known that robust Huber criterion function can handle non-Gaussian innovations/errors, but it remains unclear in the paper how is it used in the proposed model to handle non-i.i.d. data, which is regarded as one major contribution if I were right.
4. Could the authors comment on the impact of the kernel selection? I see in the experiments only a Gaussian kernel (which is a stationary kernel) is demonstrated. Don't you need a non-stationary kernel instead?
5. Experimental validation is somewhat insufficient. The selected examples (A&B) well match the linear additive model assumption. It would be interesting to see some model mismatching examples and the performance thereof. Besides, in both examples, the innovations/errors are i.i.d., right?
6. As benchmarks for comparison, only a simple AR model is adopted. The authors should compare the proposed model with more state-of-the-art models, such as LSTM, Gaussian process (GP) regression, and even some latest attention-based models. Actually, your model is sort of similar to a GP model when replacing Huber loss back to L2 loss.
7. Code is not made publicly available.
8. Many typos and grammatical errors.


**Summary Of The Paper:**

The authors proposed a robust Huber additive model for non-stationary time series prediction. They combine the idea of robust Huber regression (against non-Gaussian innovations) with a linear additive model formulated with the aid of the representer theorem in RKHS; sparsity is imposed on the weights, and the overall weights optimization problem can be solved efficiently through the classic FISTA algorithm due to the special problem structure. The major contributions lie in a robust (due to the Huber loss function) and interpretable (due to the use of linear additive model) time series prediction model as well as some theoretical supports on the performance.

**Summary Of The Review:**

To conclude, I find the paper interesting and easy to follow, the technical contributions are fine but obviously need further refinement. The literature review is insufficient, and some SOTA methods, such as LSTM, GPR, etc., are missing. The section of experimental results can be improved.

---

> ### Author Response · Authors · 2021-11-18
> **To Reviewer 4Rek (3/3)**
>
> **Q6**: Code is not made publicly available.
>
> **A6**: Thanks. The Source code package is expected to released shortly after approved by our institution.
>
> **Q7**:  Some typos
>
> **A7**:
> Thanks a lot for your careful observations. These errors have been corrected. We have paid more attention to check the spelling and grammar of the whole manuscript, e.g.,
>
> (page 1,  1st paragraph last sentence): “However” --> “In addition”
>
> (page 1,  2nd paragraph 2nd sentence): "Qiu et al. (2015) develops" --> "Qiu et al. (2015) develop"
>
> (page 2, at the 4th line in Theoretical guarantees): “With properly selected scale parameter”--> “With properly selected scale parameters”
>
> (page 2, Robust forecasting vs…): “there has been a lot of studies on …” --> “there have been a lot of studies on…”
>
> (page 2, last line)      “TS-SpA” -->  “TS-SpAM”
>
> (page 2, last line)  “neural net” --> “neural network”
>
> (Page 2, Related works a) ): "play an important role" --> "play important roles"
>
> (page 3, 1st paragraph) : “For instance, the case $X^t$ contains…”-->“For instance, the case $X^t$ that contains…”
>
> (Page 3,  Definition 1): "the most existing methods to learn" --> "the most existing methods that learn"
>
> ......
>
> **References**:
>
> [Stone 1985] C.J.  Stone. Additive regression and other nonparametric models. The Annals of Statistics. 13:689–705, 1985.
>
> [Stone 1994] C.J.  Stone. The use of polynomial splines and their tensor products in multivariate function estimation. The Annals of Statistics. 22:118–184, 1994.
>
> [Yu & Reppert 2002] Y. Yu and D. Ruppert   Penalized spline estimation for partially linear single-index models. Journal Of The American Statistical Association. 97:1042–1054, 2002.
>
> [Q. Wu, et al. 2006] Q. Wu, Y. Ying and D.X. Zhou. Learning Rates of Least-square Regularized Regression. Foundations of Computational Mathematics, 6(2):171-19, 2006.
>
> [Wasserman 2006] L. Wasserman, “All of Nonparametric Statistics,” Springer-Verlag New York, Inc., 2006.
>
> [Horowitz & Mammen 2007] J.L. Horowitz and E. Mammen. Rate-optimal estimation for a general class of nonparametric regression models with unknown link functions. The Annals of Statistics. 35(6): 2589-2619,  2007.
>
> [Yin et al. 2012] J. Yin, X. Chen and E.P. Xing. Group Sparse Additive Models. International Conference on Machine Learning, 2012.
>
> [Scott 2015] D. W. Scott, “Multivariate density estimation: theory, practice, and visualization,” John Wiley and Sons, 2015.
>
> [Lin and Zhang 2016] Y. Lin and H.H. Zhang. Component selection and smoothing in multivariate nonparametric regression. The Annals of Statistics, 34(5), 2272-2297, 2006.
>
> [Lin and Zhang 2016] Y. Lin and H.H. Zhang. Component selection and smoothing in multivariate nonparametric regression. The Annals of Statistics, 34(5), 2272-2297, 2006.
>
> [Chen 2017] Y. C. Chen, “A Tutorial on Kernel Density Estimation and Recent Advances,” arXiv: Methodology, 2017.
>
> [Bauer & Kohler 2019] B. Bauer and M. Kohler. On deep learning as a remedy for the curse of dimensionality in nonparametric regression. The Annals of Statistics.  47 (4): 2261 - 2285,  2019.
>
> [Raskutti et al. 2019] G. Raskutti, B. Yu and M.J. Wainwright. Lower bounds on minimax rates for nonparametric regression with additive sparsity and smoothness. NIPS, 2019.
>
> [Agarwal et al. 2020] R. Agarwal, N. Frosst, X. Zhang, R. Caruana, and G. E. Hinton. Neural additive models: Interpretable machine learning with neural nets. arXiv:2004.13912v1, 2020.
>
> [Wong et al. 2020] K.C Wong, Z. Li and A. Tewari. Lasso guarantees for $\beta$-mixing heavy-tailed time series. Ann. Statist. 48 (2): 1124 - 1142,  2020.

---

> ### Author Response · Authors · 2021-11-18
> **To Reviewer 4Rek (2/3)**
>
> **Q3**: It is well known that robust Huber criterion function can handle non-Gaussian innovations/errors, but it remains unclear in the paper how is it used in the proposed model to handle non-i.i.d. data, which is regarded as one major contribution if I were right
>
> **A3**: Thanks. The usage of the proposed method in handling non-i.i.d. data is as follows.
>
> In this paper, non-i.i.d. data refers to that the time series data is non-stationary or dependent.
>
> In order to handle non-stationary time series data, we propose an adaptive SpHAM based on Theorem 2. The performance of adaptive SpHAM on non-stationary data is verified from both theoretical and practical sides (the asymptotic analysis in Theorem 3 and empirical evaluations  in Section 4).
>
> Moreover, the asymptotically consistent upper bounds (Theorems 1 and 3) indicate that our method can handle non-independent time series data, since all the theoretical findings are given without the assumption of data independence. Moreover, note that without the asymptotic dependence conditions, one may get slower convergence rates (e.g, $T^{-1/2}$ in this paper) indicating the sacrifice for learning non-independent data that may exist overlapping information.
>
> **Q4**: Could the authors comment on the impact of the kernel selection? I see in the experiments only a Gaussian kernel (which is a stationary kernel) is demonstrated. Don't you need a non-stationary kernel instead?
>
> **A4**: Thanks for your valued comment. As far as we know, the choice of kernel function does not play an important role in kernel density estimation (KDE) for the constructing reproduced kernel Hilbert space (RKHS), as shown in Chen [2017]. The effect of the kernel function on the estimation error is just a constant shift (see equation (2) in Chen [2017]), and the difference is generally very small among common kernel functions (see, e.g., page 72 of [Wasserman 2006] and Section 6.2.3 in [Scott 2015]).
>
> **Q5**: Experimental validation is somewhat insufficient. The selected examples (A&B) well match the linear additive model assumption. It would be interesting to see some model mismatching examples and the performance thereof. Besides, in both examples, the innovations/errors are i.i.d., right? As benchmarks for comparison, only a simple AR model is adopted. The authors should compare the proposed model with more state-of-the-art models, such as LSTM, Gaussian process (GP) regression, and even some latest attention-based models. Actually, your model is sort of similar to a GP model when replacing Huber loss back to L2 loss.
>
> **A5**: In both examples, the innovations/errors are i.i.d, which is consistent with the data generation model (1).
>
> Following your suggestion, we strengthen our empirical evaluations from the following aspects:
>
> a) Add a mismatching example and verify the promising performance of our extended method; see A1 for more details.
> b) Add more discussion on the experimental results, e.g., the investigation of the impact of the hyper-parameter (e.g., $\sigma$ ) on the performance of our method.
> c) Add more comparison experiments, such as LSTM and Gaussian processes (GP) regression.

---

> ### Author Response · Authors · 2021-11-18
> **To Reviewer 4Rek (1/3)**
>
> Thank you for your thorough review and constructive comments. All your concerns have been carefully addressed as below. The manuscript is carefully revised accordingly. We sincerely hope our responses fully address your questions.
>
> **Q1**:	Although the simple linear additive model is well interpretable, its modeling capacity is quite weak on the other hand. As a remedy not destroying your optimization framework, you may consider adding some interacting terms, like $Y^2*Y^3$.
>
> **A1**:  Thanks for your valuable suggestion. We respectfully note that linear additive model can be applied combining deep neural networks, as shown in [Agarwal et al. 2020] [ Bauer & Kohler 2019], which ensures a significantly large modeling.
>
> Moreover, the functions containing $Y^2*Y^3$ (or other interaction terms) can be fitted by our newly added group additive model, group SpHAM, an extension from the linear additive model, which divides the input space into $d$ subgroups (see Remark 1 for more details).
>
> Following your valuable suggestion, we conduct an addition experiment. The data is generated according to $Y^t = sin(Y^{t-1}*Y^{t-2})$.  Group SpHAM is then trained on the data, where the input space is divided into 3 subgroups (i.e., ${\mathcal{X}_1}$, ${\mathcal{X}_2}$ and $(\mathcal{X}_1, \mathcal{X}_2)$).  The results are summarized in the following table:
>
> |  | Gaussian Noise |  | Student Noise |
> |---|---|---|---|
> | Method | ASE       TD |  | ASE       TD |
> | AR | 0.392    0.206 |  | 2.371   0.206 |
> | GPR | 0.224    0.113     |  | 1.389   0.303   |
> | LSTM | 0.330    0.141 |  | **1.023**   0.398  |
> | TS-SpAM | **0.228**    0.128 |  | 1.396   0.401 |
> | SpHAM | 0.280    0.208 |  | 1.351   0.342 |
> | Group SpHAM | 0.239    **0.125** |  | 1.344   **0.288** |
>
> This table verifies the effectiveness of our group SpHAM. This has been added to Section 4.1.
>
> **Q2**:  The proposed robust time series prediction model with the three theorems constitutes the main contribution of the paper. But to me, the meanings of the lower bound (namely each part of the right-hand side) are hard to interpret. Are they tight bounds in general? Besides, are f solutions of the corresponding optimization problems?
>
> **A2**：Thanks. With different data assumptions, these three Theorems give the upper bound of the function approximation error between our estimator and the ground truth $f^*$, respectively:
>
> a) The bound in Theorem 1 applies to stationary time series data. This bound depends on the training sample size $T$, which indicates that our estimator is asymptotically consistent with the ground truth when the sample size $T\rightarrow \infty$. This bound is significantly different from the previous robust time series models [Qiu et al. 2015][Wong et al. 2020], since we allow time series data to be dependent on history (most previous works are established under asymptotic independent data assumption, e.g., various mixing conditions).
>
> b) The bounds in Theorems 2-3 apply to non-stationary time series. Compared with Theorem 1, the upper bound in Theorem 2 contains an interesting term disc($\mathbf{s}$) \geq 0, which characterizes the discrepancy between the empirical distribution of the sample and target distribution that we aim to predict. Theorem 2 indicates that, even if the sample size $T$ tends to infinity, the bound may not tend to be 0 (because disc($\mathbf{s}$) is not equal to 0 when the time series data is non-stationary). Inspired by Theorem 2, we further propose an adaptive SpHAM (see Section 3.3) by penalizing the disc($\mathbf{s}$) to handle non-stationary data. Under some mild assumptions, the bound of this new adaptive SpHAM converges to 0 when the sample size $T$ tends to infinity, as shown in Theorem 3. In this way, Theorem 3 characterizes the asymptotic property of our proposed adaptive SpHAM.
>
> A lower bound is needed to verify whether the upper bound is tight: if the lower bound and upper bound are of the same order, they are both tight. We are not able to verify whether our bounds are tight without the lower bound.  Indeed, the lower bound analysis appears in a few studies on additive models under i.i.d data assumption, see, e.g., [Raskutti et al. 2019]. Based on the analysis technique of lower bound on additive models [Raskutti et al. 2019], it is a very interesting and meaningful future direction to obtain the tight bound of our SpHAM for non-i.i.d data.
>
> The solutions of our optimization algorithm is $\hat{f}$. Since the optimization problem is strongly convex (convex Huber loss + convex regularizer). Gradient-based optimization can thus find the optimal solution $\hat{f}$.

---

> > ### Comment · Reviewer_4Rek · 2021-11-22
> > **Thanks for your responses**
> >
> > First, I would like to thank the authors for providing quite detailed responses, which answered most of my questions. I also read other reviewers' comments and your responses carefully.
> >
> > I don't deny that this paper presents some solid results. The considered problem is of good value, but the theoretical results, being clean and solid, are not very useful. Some of the important questions from Reviewers 1&2 are not well answered and left for future work. Concerning the newly added results, I doubt that the performance of the proposed method might be comparable with GPR model using a more advanced non-stationary kernel; see S. Kaski's papers, for instance. Code is still not available.
> >
> > Overall, I like the theoretical contributions of this work. I will likely increase the mark to 6, although it somehow locates between 5 and 6.

---

> > > ### Author Response · Authors · 2021-11-26
> > > **Thanks for your suggestions and support! (2/2)**
> > >
> > > **Q4**: Code is still not available
> > >
> > > **A4**: Thanks. We are associated with an industrial organization. Releasing code needs approval of our institution. We have submitted our applications. The Source code package will be released once the application is approved.
> > >
> > > **References**:
> > >
> > > [Rahimi & Benjamin 2007] A. Rahimi and R. Benjamin. Random Features for Large-Scale Kernel Machines. NIPS, 2007.
> > >
> > > [Wang & Zhou 2011] C. Wang, D.X. Zhou, Optimal learning rates for least squares regularized regression with unbounded sampling, Journal of Complexity. 27:55–67, 2011.
> > >
> > > [Heinonen et al. 2016] Markus Heinonen, Henrik Mannerström, Juho Rousu, Samuel Kaski, Harri Lähdesmäki. Non-Stationary Gaussian Process Regression with Hamiltonian Monte Carlo. PMLR, 51:732-740, 2016.
> > >
> > > [Raskutti et al. 2019] G. Raskutti, B. Yu and M.J. Wainwright. Lower bounds on minimax rates for nonparametric regression with additive sparsity and smoothness. NIPS, 2019
> > >
> > > [Feng and Wu 2020] Yunlong   Feng   and   Qiang   Wu. A   statistical   learning   assessment   of   huber   regression. arXiv:2009.12755v1, 2020.

---

> > > ### Author Response · Authors · 2021-11-26
> > > **Thanks for your suggestions and support! (1/2)**
> > >
> > > Thank you very much for your helpful suggestions and kind support!
> > >
> > > **Q1**. The theoretical results, being clean and solid, are not very useful.
> > >
> > > **A1**: Thanks. We respectfully note that the theoretical results are crucial to the construction of our adaptive SpHAM. Specifically, the optimization problems (9) and (10) of the adaptive SphAM are formed by combining the ingredients from the upper bounds in Theorem 3. With the theory-inspired optimization problems, the approximation ability of the estimator is well established from the theoretical aspect. Moreover, the techniques used in our theoretical analysis may be of independent interest in designing other robust methods for non-stationary time series analysis, since it stands for a large class of loss functions with Lipschitz property (e.g., the error decomposition in Prop. 1 and the bounds on sample error, hypothesis error, approximation error in Props. 2-4).
> > >
> > > **Q2**: I doubt that the performance of the proposed method might be comparable with GPR model using a more advanced non-stationary kernel;
> > >
> > > **A2**: Thanks for your valuable question. We agree that GPR with non-stationary kernel [Heinonen et al. 2016] achieves comparable performance under some scenarios, e.g., the nonstationary situation with input-dependent variance Gaussian innovation. However, our method has the following advantages:
> > >
> > > **1) Our method applies to the zero-mean asymmetric innovation distribution**: According to [Feng and Wu 2020], our method applies to asymmetric zero-mean distributions of innovation.  In order to verify this, we conduct an additional experiment, which is in full agreement with our claims.
> > >
> > > In the experiment, we consider the Example B (this example can better reflect the influence of the skewed innovation, because GPR and our method are comparable under Gaussian and Student innovation under this case), where the innovations are drawn from a skewed distribution with zero-mean and 0.96 skewness (generated by the skewnorm python package with parameter $\alpha=10$).  We implement GPR by the MATLAB toolbox provided by [Heinonen et al. 2016]. The evaluations are repeated for 50 times. The following average results show the promising performance of our group SpHAM.
> > >
> > > |           |             | Skewed distribution|        |
> > > |-----------|-------------|----------|--------|
> > > |   | Method | ASE        |        TD |
> > > | Example B | GPR         | 0.6958   | 0.2948 |
> > > |           | Group SpHAM | 0.7007   | 0.2791 |
> > >
> > >
> > > **2) Our method is more interpretable**: The additive structure and sparsity-inducing regularizer offer our method interpretable results, e.g., the Granger causal detection in our real-world experiments.
> > >
> > >
> > >
> > > **Q3**:  Some of the important questions from Reviewers 1&2 are not well answered and left for future work.
> > >
> > > **A3**: Thanks. In this paper, we have made significant contributions as follows:
> > >
> > > (1)  We propose novel robust methods, SpHAM and adaptive SpHAM, with solid theoretical guarantees for (non)stationary time series forecasting.  The theoretical results are established without explicit stationary and independent data assumption, compared to existing works. To our best knowledge, there does not exist sharper bounds under the same conditions.
> > >
> > > (2) Both the proposed SpHAM and adaptive SpHAM are implemented via an effective learning algorithm, Algorithm 1.
> > >
> > > (3) Comprehensive empirical evaluations are conducted to verify the effectiveness of our method, which fully support our algorithms.
> > >
> > > We sincerely appreciate all the valuable suggestions from the reviewers. Besides more detailed discussions, we will further add the random Fourier features technique (discussed in A3 to Reviewer 3c4P) accordingly to the revised manuscript:
> > >
> > > The random Fourier features technique is a plug-and-play method [Rahimi & Benjamin 2007], which can easily apply to our Algorithm 1 by introducing the Fourier transform associated with the kernel $K_j$.  The computational time of Algorithm 1 equipped with random Fourier features technique will be shortened significantly in the presence of large-scale data.

---

### Official Review · Reviewer_es4A · 2021-11-03

**Correctness:** 3
**Technical Novelty And Significance:** 2
**Empirical Novelty And Significance:** Not applicable
**Recommendation:** 6
**Confidence:** 2

**Main Review:**

The novelty of this ms is mainly theoretical. It introduces a non-Gaussian noise to the additive model can come up with the error bounds. The experiments are not comprehensive. It only compares with two other auto regressive methods. I will be interested to see its performance in non-autoregressive setting, i.e. simply sum several non-linear functions and Gaussian / non-Gaussian noise and see if the ground truth could be recovered. Also, the real data experiment (F.1) result is only a figure without any interpretation. It will be nice to see if this method provides new insights compared with previous methods.

**Summary Of The Paper:**

This ms tackles a timer series problem where each observation is the sum of p hidden functions and a noise (termed as innovation in ms). Traditionally the noise is Gaussian, but this ms makes the noise into heavy tailed distribution such as t distribution. The L2 loss function of the original problem is then replaced by so called Huber loss, which introduces a cutoff to the deviations and is less affected by outliers than L2 loss. Based on this new loss function, this ms sets up the sparse Huber additive model and derive the bounds of the learned hidden functions, under stationary processes. In case of non-stationary processes, the ms introduces different weights to each time points, which can be used to calculate a discrepancy that reflects the non-stationarity. With different weights to different time points, the ms come up with new error bounds. Inference algorithms are provided in the end.

**Summary Of The Review:**

My general feeling is that there is theoretical contribution but the experiments could be improved.

---

> ### Author Response · Authors · 2021-11-18
> **To Reviewer  es4A**
>
> Thank you for your thorough review and constructive comments. All your concerns have been carefully addressed as below. The manuscript is carefully revised accordingly. We sincerely hope our responses fully address your questions.
>
> **Q1**: I will be interested to see its performance in non-autoregressive setting, i.e. simply sum several non-linear functions and Gaussian / non-Gaussian noise and see if the ground truth could be recovered. Also, the real data experiment (F.1) result is only a figure without any interpretation. It will be nice to see if this method provides new insights compared with previous methods.
>
> **A1**:  Thanks for your valuable question. As described in Remark 1, our SpHAM can be easily extended to the group SpHAM by replacing the direct decomposition $\mathcal{X}_j, j=1,...,p $ with the subgroups decomposition $\mathcal{X}_i, i=1,...,d$, where each $\mathcal{X}_i$ corresponds to the component input space concerning the interactions among variables.  Group SpHAM, as a robust version of Group SpAM [Yin et al. 2012] and COSSO [Lin and Zhang 2016], can better deal with complex data-generating model, e.g., the data-generating model contains interacting items, instead of simply summing several non-linear functions. To verify our claim, we conducted an additional empirical evaluation, where the data is generated by $Y^t = sin(Y^{t-1}*Y^{t-2})$. The results are summarized as below:
>
> |  | Gaussian Noise |  | Student Noise |
> |---|---|---|---|
> | Method | ASE       TD |  | ASE       TD |
> | AR | 0.392    0.206 |  | 2.371   0.206 |
> | GPR | 0.224    0.113     |  | 1.389   0.303   |
> | LSTM | 0.330    0.141 |  | **1.023**   0.398  |
> | TS-SpAM | **0.228**    0.128 |  | 1.396   0.401 |
> | SpHAM | 0.280    0.208 |  | 1.351   0.342 |
> | Group SpHAM | 0.239    **0.125** |  | 1.344   **0.288** |
>
>
> The results verify the promising modeling capacity of our group SpHAM for complex time series data.
>
> Beside, we also strengthen our empirical evaluations from the following aspects:
>
> a)	We have conducted an additional experiment on Coronal Mass Ejections (CMEs) data in the revised manuscript (Appendix F.4). CMEs is collected from the Richardson and Cane List (http://www.srl.caltech.edu/ACE/ASC/DATA/level3/icmetable2.htm), SOHO LASCO CME Catalog (https://cdaw.gsfc. nasa.gov/CME_list/), and OMNIWeb Plus (https://omniweb.gsfc.nasa.gov/). We select all 152 observations from 1996 to 2016. A total of 9 features of CMEs, including Angular Width, MASS, and CMEs arrival time (see our revised paper for more details).  Some interesting findings are concluded; e.g., our inferred Granger causal network shows the CMEs Angular Width does not cause the CMEs arrival time forecasting, while [Liu et. al. 2018] suggests that they have a significant correlation. This indicates that the CMEs arrival time is affected by the Angular Width at current time, but not by the historical Angular Width.
>
> b)	We have added more discussions on the experimental results, e.g., the interpretations for Fig. 3.
>
> c)	We have also investigated the impact of the hyper-parameter (e.g., $\sigma$) on the performance of our method. For fixed sample size $T=400$, the empirical results over different $\sigma=T^\alpha$ are collected below:
>
> | Example 1 | $t$ noise |  |  |  |  |
> |---|---|---|---|---|---|
> | $\sigma$ | $T^{16}$ | $T^{4}$ | $T^{1/4}$ | $T^{1/8}$ | $T^{1/24}$ |
> | ASE | 1.34 | 1.19 | 1.04 | 0.98 | 0.98 |
> | TD | 1.02 | 0.63 | 0.51 | 0.42 | 0.44 |
>
>
> This table shows that, for heavy-tailed $t$ noise, our method significantly out-perform with smaller $\alpha$, which validates Theorem 1 and Fig 1.
>
> d) We add more comparison methods, following your suggestions, such as LSTM and Gaussian processes (GP) regression.
>
> e) In order to make the results more stable, we repeat all our experiments for 100 times.
>
> Some of the results (e.g., Example A) are shown as follows (all will be presented in our revised paper due to space limitations):
>
> |  |  | Gaussian Noise  |  | Student Noise |
> |---|---|---|---|---|
> | Methods |  |  ASE       TD |  |  ASE      TD |
> | AR |  | 1.220     1.225 |  | 2.007   1.304 |
> | GPR  |  | 0.710     0.712 |  | 1.203   0.807 |
> | LSTM |  | 0.271     0.267 |  | 1.694   0.833 |
> | TS-SpAM |  | 0.116     0.105 |  | 1.154   0.604 |
> | SpHAM |  | 0.118     0.105 |  | 1.001   0.484 |
>
> **References**:
>
> [Lin and Zhang 2016] Y. Lin and H.H. Zhang. Component selection and smoothing in multivariate nonparametric regression. The Annals of Statistics, 34(5), 2272-2297, 2006.
>
> [Yin et al. 2012] J. Yin, X. Chen and E.P. Xing. Group Sparse Additive Models. International Conference on Machine Learning, 2012.
>
> [ Liu et al. 2018] J. Liu, Y. Ye, C. Shen, Y. Wang and R. Erdelyi. A New Tool for CME Arrival Time Prediction using Machine Learning Algorithms: CAT-PUMA. The Astrophysical Journal. 855. 2018

---

### Official Review · Reviewer_KeVD · 2021-11-03

**Correctness:** 3
**Technical Novelty And Significance:** 3
**Empirical Novelty And Significance:** 2
**Recommendation:** 6
**Confidence:** 3

**Main Review:**

- Assumption 1 on boundedness of $|Y^t|$ is very strong and may not hold even for simple distributions such as Gaussian. The authors are encouraged to relax this assumption, otherwise the results of the manuscript will be limited and not general enough.

- Authors need to either verify or give references to why certain kernels would satisfy Assumption 2.

- The authors claim in the abstract that `"... we propose an adaptive sparse Huber additive model for robust
forecasting and inference (e.g., Granger causal discovery) …". However, (a) there is no results about inference in the manuscript which would quantify the uncertainty of estimation; (b) The Granger causal network is only mentioned in the appendix for a real data set while it is not discussed how this network is estimated using the proposed algorithm and whether the estimated network makes sense scientifically. The authors need to add results related to inference, or at least modify the abstract accordingly.

- The upper bound in Theorem 2 is unusual. It is appropriate for the authors to interpret the results of this theorem. Also, are the results sharp?


- There are several hyper-parameters in the proposed algorithm, and as mentioned at top of page 9, the authors did not attempt to find data-driven methods to select such parameters. In is not clear whether the proposed methodology would still work well under different simulation settings using the same fixed hyper-parameters. A better treatment of hyper-parameter selection is needed for the method to become practical.

- Experiments in Section 4.1 are repeated only 10 times which is quite low. These studies need to be replicated more, say 100 times.

- The code for reproducing empirical results is not provided.




- The sentence above Definition 2 in page 7 needs rewriting.

- Statement of Theorem 3 needs rewriting: "Let Assumption 4 the conditions in Theorem 2 be true".


- Typo at the beginning of section 4.2 on page 9: "We test our algorithm on nonlinear Nonlinear-VAR dataset".



**Summary Of The Paper:**

In this manuscript, the authors proposed an adaptive sparse Huber additive model and provide an algorithm to handle non-parametric estimation of the underlying unknown function. Theoretical guarantees are provided under certain conditions and some empirical comparison with competing methods are provided.

**Summary Of The Review:**

- A more comprehensive discussion on Assumptions are needed.
- More interpretations about the theoretical results need to be provided.
- Some claims in the abstract need to be modified.
- A better treatment of hyper-parameter selection is required.
- Some typos need to be fixed.

---

> ### Author Response · Authors · 2021-11-18
> **To Reviewer KeVD (3/3)**
>
> **Q5**:  There are several hyper-parameters in the proposed algorithm, and as mentioned at top of page 9, the authors did not attempt to find data-driven methods to select such parameters. In is not clear whether the proposed methodology would still work well under different simulation settings using the same fixed hyper-parameters. A better treatment of hyper-parameter selection is needed for the method to become practical. Experiments in Section 4.1 are repeated only 10 times which is quite low. These studies need to be replicated more, say 100 times.
>
> **A5**: Thanks. We apply grid search via cross validation to select the hyper-parameters, including the $\sigma$ of Huber loss, the bandwidth of $RKHS$, and other regularization parameters. The search space is as follows: $\sigma=T^\alpha, \alpha\in\{1/2, 1/4, 1/8, 1/16, 1/32, 1/48\}$, the bandwidth of RHKS $d\in\{0.1, 0.5, 1, 5\}$, and the regularization parameter $\lambda \in\{0.0001, 0.0005, 0.001, 0.005, 0.01, 0.05, 0.1, 0.5, 1, 5\}$. Note that the hyper-parameter selection approach for $\sigma=T^\alpha$ is data-driven. We have also investigated the impact of the selection of $\sigma$ on our method. For the fixed sample size $T=400$, the empirical results of our SpHAM over different $\sigma=T^\alpha$ are summarized as below:
>
> | Example 1 | $t$ noise |  |  |  |  |
> |---|---|---|---|---|---|
> | $\sigma$ | $T^{16}$ | $T^{4}$ | $T^{1/4}$ | $T^{1/8}$ | $T^{1/24}$ |
> | ASE | 1.34 | 1.19 | 1.04 | 0.98 | 0.98 |
> | TD | 1.02 | 0.63 | 0.51 | 0.42 | 0.44 |
>
> This table shows that, for heavy-tailed $t$ noise, our method tends to perform better with smaller $\alpha$. This result validates the Theorem 1 and Fig 1.
> Following your suggestion, we have repeated all our experiments for 100 times. Some of the results (e.g., Example A) are shown as follows (all will be presented in our revised paper due to space limitations):
>
> |  |  | Gaussian Noise  |   | Student Noise |
> |---|---|---|---|---|
> | Methods |  |  ASE       TD |  |  ASE      TD |
> | AR |  | 1.220    1.225 |  | 2.007   1.304 |
> | GPR  |  | 0.710    0.712 |  | 1.203   0.807 |
> | LSTM |  | 0.271     0.267 |  | 1.694   0.833 |
> | TS-SpAM |  | **0.116**    **0.105** |  | 1.154   0.604 |
> | SpHAM |  | 0.118     **0.105** |  | **1.001**  **0.484** |
>
> **Q6**:	The code for reproducing empirical results is not provided.
>
> **A6**: Thanks. The source code package is expected to be released publicly shortly after approval of our institution.
>
> **Q7**: Some typos
>
> **A7**: Thanks and addressed.
>
> **References**:
>
> [Q. Wu, et al. 2006] Q. Wu, Y. Ying and D.X. Zhou. Learning Rates of Least-square Regularized Regression. Foundations of Computational Mathematics, 6(2):171-19, 2006.
>
> [Smale & Zhuo 2007] S. Smale, D.X. Zhou. Learning Theory Estimates via Integral Operators and Their Approximations. Constructive Approximation. 26:153–172, 2007.
>
> [Caponnetto & DeVito 2007] A. Caponnetto and E. DeVito, Optimal rates for regularized least-squares algorithms. Foundations Of Computational Mathematics.7: 331–368, 2007.
>
> [Wu & Zhou 2008] Q. Wu and D.X. Zhou. Learning with sample dependent hypothesis spaces. Computers and Mathematics with Applications. 56:2896–2907, 2008.
>
> [Steinwart & Christmann 2008] I. Steinwart, A. Christmann, Support Vector Machines, Springer Science and Business Media, 2008.
>
> [Sun & Wu 2011] H. Sun and Q. Wu. Least square regression with indefinite kernels and coefficient regularization. Applied and Computational Harmonic Analysis. 30:96-109, 2011.
>
> [Wang & Zhou 2011] C. Wang, D.X. Zhou, Optimal learning rates for least squares regularized regression with unbounded sampling, Journal of Complexity. 27:55–67, 2011.
>
> [Qiu et al. 2015] H. Qiu, S. Xu, F. Han, H. Liu, and B. Caffo. Robust estimation of transition matrices in high dimensional heavy-tailed vector autoregressive processes. In Proceedings of the 32nd International Conference on Machine Learning, volume 37, pp. 1843–1851, 2015.
>
> [S. Lin et al. 2017] S. Lin, X. Guo, D.X. Zhou. Distributed Learning with Regularized Least Squares. Journal of Machine Learning Research, 18(92):1−31, 2017.
>
> [Y. Yang et al. 2018] Y. Yang, A.W. Yu, Z. Wang and T. Zhao. Detecting nonlinear causality in multivariate time series with sparse additive models. arXiv:1803.03919v2, 2018.
>
> [Liu et. al. 2018] J. Liu, Y. Ye, C. Shen, Y. Wang and R. Erdelyi. A New Tool for CME Arrival Time Prediction using Machine Learning Algorithms: CAT-PUMA. The Astrophysical Journal. 855(2): 109. 2018.
>
> [Wong et al. 2020] K.C Wong, Z. Li and A. Tewari. Lasso guarantees for $\beta$-mixing heavy-tailed time series. Ann. Statist. 48 (2): 1124 - 1142,  2020.
>
> [Kuznetsov&Mohri 2020] V. Kuznetsov and M. Mohri. Discrepancy-based theory and algorithms for forecasting non-stationary time series. Annals of Mathematics and Artificial Intelligence, 88:367–399, 2020.

---

> ### Author Response · Authors · 2021-11-18
> **To Reviewer KeVD (2/3)**
>
> **Q3**: The authors claim in the abstract that `"... we propose an adaptive sparse Huber additive model for robust forecasting and inference (e.g., Granger causal discovery) …". However, (a) there is no results about inference in the manuscript which would quantify the uncertainty of estimation; () (b) The Granger causal network is only mentioned in the appendix for a real data set while it is not discussed how this network is estimated using the proposed algorithm and whether the estimated network makes sense scientifically. The authors need to add results related to inference, or at least modify the abstract accordingly.
>
> **A3**:  Thanks. Following your suggestion, we will modify the abstract accordingly to highlight our focus in this paper (i.e., robust forecasting model for non-Gaussian and nonstationary time series with theoretical guarantees).  Moreover, we will add more interpretations for the Granger causal network in Fig. 3. To further verify the inference ability of our method, we also conduct an additional experiment on Coronal Mass Ejections (CMEs) data in the revised manuscript (Appendix F.4). CMEs is collected from the Richardson and Cane List (http://www.srl.caltech.edu/ACE/ASC/DATA/level3/ icmetable2.htm), SOHO LASCO CME Catalog (https://cdaw.gsfc. nasa.gov/CME_list/), and OMNIWeb Plus (https://omniweb.gsfc.nasa.gov/). We select all 152 observations from 1996 to 2016. A total of 9 features of CMEs, including Angular Width, MASS, and CMEs arrival time (see our revised paper for more details).  Some interesting findings are concluded; e.g., our inferred Granger causal network shows the CMEs Angular Width does not cause the CMEs arrival time forecasting, while [Liu et. al. 2018] suggests that they have a significant correlation. This indicates that the CMEs arrival time is affected by the Angular Width at current time, but not by the historical Angular Width.
>
> **Q4**: The upper bound in Theorem 2 is unusual. It is appropriate for the authors to interpret the results of this theorem. Also, are the results sharp?
>
> **A4**: Thank. We will add a detailed explanation as below.
>
> The upper bound in Theorem 2, compared with Theorem 1, contains an interesting term disc($\mathbf{s}$), which characterizes the discrepancy between the empirical distribution of the sample and the target distribution that we aim to predict. Particularly, when the time series is stationary, disc(\mathbf{s}) equals 0, and thus Theorem 2 reduces to Theorem 1. Otherwise, the upper bound is not necessarily 0, even as $T\rightarrow \infty$. Theorem 2 inspires our adaptive SpHAM (by penalizing disc($\mathbf{s}$); see Section 3.3) for handling non-stationary time series data.
>
> Recall that the derived convergence rate $O(T^{-1/2})$ in Corollary 1, which is competitive with the recent robust time series models [Qiu et al. 2015][Wong et al. 2020]. Moreover, our method is novel because we do not require explicit assumptions on the data dependency (e.g., various mixing conditions).  As far as we know, we are amongst the sharpest bounds (e.g., no explicit assumption on the data dependency). Motivated by your question, we have an idea to obtain a sharper bound by utilizing Sequential covering number [Kuznetsov&Mohri 2020]. This is a very interesting future direction.

---

> ### Author Response · Authors · 2021-11-18
> **To Reviewer KeVD (1/3)**
>
> Thank you for your thorough review and constructive comments. All your concerns have been carefully addressed as below. The manuscript is carefully revised accordingly. We sincerely hope our responses fully address your questions.
>
> **Q1**:	Assumption 1 on boundedness of $|Y^t|$ is very strong and may not hold even for simple distributions such as Gaussian. The authors are encouraged to relax this assumption, otherwise the results of the manuscript will be limited and not general enough.
>
> **A1**:  Thanks for your valuable suggestion. This assumption stands when the distribution of $|Y^t|$ is truncated, such as truncated Gaussian. This assumption is important to characterize the capacity of the data-dependent hypothesis space containing $\hat{f}$, which holds in the real-world scenario where the data is of finite value. Also, this assumption has been wildly used in the learning theory community; please refer to [Q. Wu, et al. 2006][Smale & Zhuo 2007][Wu & Zhou 2008][Sun & Wu 2011][S. Lin et al. 2017].  This has been carefully discussed in the revised manuscript (see Section 3.2.1; page 5). Moreover, following your valuable suggestion, future work will focus on relaxing Assumption 1 with moment hypothesis concerning unbounded outputs in [Wang & Zhou 2011][Caponnetto & DeVito 2007].
>
> **Q2**: Authors need to either verify or give references to why certain kernels would satisfy Assumption 2.
>
> **A2**: Thanks. Assumption 2 only requires that the kernel is bounded for any $x \in \mathcal{X}$. Recalling the space $\mathcal{X}$ is assumed to be compact (as assumed in many existing works), Assumption 2 holds for all Mercer kernels (e.g., Gaussian kernel), which has been verified in many literatures; see, e.g., [Q. Wu, et al. 2006][Wu & Zhou 2008][Steinwart & Christmann 2008]. Following your suggestion, we have added detailed explanations and references in our revised manuscript (see Section 3.2.1; page 5).

---

### Official Review · Reviewer_3C4P · 2021-11-06

**Correctness:** 3
**Technical Novelty And Significance:** 3
**Empirical Novelty And Significance:** 3
**Recommendation:** 8
**Confidence:** 2

**Main Review:**

Overall, I thought this was a solid paper.  The SpHAM model is not that novel, but the analysis of the model appeared novel, correct and of interest to those working with real world data (e.g. financial data) that is non i.i.d., non-stationary and heavy tailed for which "practical" theoretical findings are not that common.

Pros:
-I think this was a well written paper. The development was clean and logical and the authors nicely illustrated their contributions.
-I think Theorem 1/Corollary 1 is a tidy result.  Expressing the rate of convergence via a moment condition, c, and the huber threshold param provides some nice intuition for situations where the method will/will not be effective.
-The experiments nicely demonstrated the utility of the SpHAM approach for heavy tailed, stationary data and the adaptive SpHAM for non-stationary data.

Cons:
-How restrictive is assumption 3 in practice? Since we don't know the true f*, it that condition testable/verifiable?
-How do we know that a q being a uniform distribution over the most recent observations satisfies Assumption 4?  Seems like if l is to large its not likely to hold, but if l is too small then we can't capture any of the non-stationarity.
-Can you provide a bit more color on how Step A/eq (9) is solved?
-Can you comment on the scalability/space, time complexity of algorithm 1?  Could the method be scaled to longer time series and do you envision an incremental/online version of the procedure?
-How does performance change if t/400 is changed in experiment B?  i.e. as the drift to noise scale ratio is changed.


**Summary Of The Paper:**

The authors propose, theoretically analyze and empirically evaluate a sparse additive huber additive model (SpHAM), which is an additive model where each component function in the model is a kernel function that is fitted by minimizing a huber huber loss function with a functional sparsity inducing penalty term.  Statistical properties when estimating this model on stationary data with non gaussian noise and also on non-stationary data are developed. Finally, the model and fitting routines are demonstrated empirically on 2 synthetic and 1 real world time series.

**Summary Of The Review:**

Overall, I think this was a good paper with some nice theoretical developments and an easily understandable and interpretable model.

---

> ### Author Response · Authors · 2021-11-18
> **To Reviewer 3C4P (2/2)**
>
> **Q4**:	 *Could the method be scaled to longer time series and do you envision an incremental/online version of the procedure?*
>
> **A4**: Thanks for your valuable question. Designing the online version of our method for longer time series is a very interesting future work! To our best knowledge, there does not exist an online version of additive time series model for robust and non-stationary forecasting. Our future work will be emphasized on investigating the regret bound of the corresponding online version of robust additive time series model, inspired by the approaches of sequential covering number/Rademacher complexity in exploring the generalizability of online learning [Rakhlin & Sridharan 2015].
>
> **Q5**: *How does performance change if t/400 is changed in experiment B? i.e. as the drift to noise scale ratio is changed.*
>
> **A5**: Thanks. This is a very interesting question. Experiment B describes a scenario of the data generating process with smooth distribution drift, which coincides with the Assumption 4. The ratio $t/400$ characterizes the degree of change in the data distribution over time $t$. Under this data-generating process, the change of this ratio does not have significant impact to the forecasting of our method. According to Theorem 3, no matter how the ratio changes, our method can still achieve satisfactory performance by finding the corresponding optimal $\mathbf{s}$.
>
> **References**:
>
> [Tao & An 1998] P.D. Tao and L.T.H An. A d.c. optimization algorithm for solving the trust-region subproblem. SIAM Journal on Optimization, 8:476–505, 1998.
>
> [Cucker & Smale 2002] F. Cucker and S. Smale. On the mathematical foundations of learning. Bulletin (New Series) of the American Mathematical Society, 39: 1-49, 2002.
>
> [Smale & Zhou 2003] S. Smale, D.X. Zhou, Estimating the approximation error in learning theory, Analysis & Applications. 1: 17–41, 2003.
>
> [Qiang Wu, et.al 2006] Q. Wu, Y. Ying and D.X. Zhou. Learning Rates of Least-square Regularized Regression. Foundations of Computational Mathematics, 6(2):171-19, 2006.
>
> [Cucker & Zhou. 2007] F. Cucker and D.X. Zhou. Learning Theory: An Approximation Theory Viewpoint, Cambridge University Press, 2007.
>
> [Colson et al. 2007] B. Colson, P. Marcotte and G. Savard. An overview of bilevel optimization. Annals of operations research, 153(1):235-56, 2007.
>
> [Rosasco et al. 2009] L. Rosasco, M. Belkin and E. De Vito.  A Note on Learning with Integral Operators. COLT, 2009.
>
> [Anava et al. 2013] O. Anava, E. Hazan, S. Mannor and O. Shamir. Online Learning for Time Series Prediction. PMLR 30:172-184, 2013.
>
> [Rakhlin & Sridharan 2015] A. Rakhlin and K. Sridharan. Online Learning via Sequential Complexities. Journal of Machine Learning Research, 16: 155-186, 2015.
>
> [Lin & Rosasco 2016] J. Lin and L. Rosasco. Optimal Learning for Multi-pass Stochastic Gradient Methods. NIPS, 2016.
>
> [Christmann & Zhou 2016] A. Christmann and D.X. Zhou. Learning rates for the risk of kernel based quantile regression estimators in additive models. Analysis and Applications, 14(3):449–477, 2016.
>
> [Ochs et al., 2016] P. Ochs, R. Ranftl, T. Brox, and T. Pock. Techniques for gradient based bilevel optimization with nonsmooth lower level problems. Journal of Mathematical Imaging and Vision, 56(2):175–194, 2016.
>
> [Kuznetsov&Mohri 2020] V. Kuznetsov and M. Mohri. Discrepancy-based theory and algorithms for forecasting non-stationary time series. Annals of Mathematics and Artificial Intelligence, 88:367–399, 2020.

---

> ### Author Response · Authors · 2021-11-18
> **To Reviewer 3C4P (1/2)**
>
> Thank you very much for your constructive comments and kind support! All your concerns have been carefully addressed as below. The manuscript is carefully revised accordingly. We sincerely hope our responses fully address your questions.
>
> **Q1**:	*How restrictive is assumption 3 in practice? Since we don't know the true $f^{\*}$, it that condition testable/verifiable?*
>
> **A1**:  Thanks. Assumption 3 stands in most practical cases.  Assumption 3 requires $f^*$ to be a function lying in the range of the $L^r_{K_j, T+1}$, where $L^r_{K_j, T+1}$ is a $r$-th power compact and positive integral operator associated with kernel $K_j$ (see the related definition  in page 5). For example, if $r=0.5$, the ground truth function $f^*$ needs to be a real-valued function in the reproducing kernel Hilbert space (RKHS). The RKHS admits the bounded and continuous real-valued functions in Hilbert space. Moreover, this assumption has been widely used in learning theory; please refer to [Cucker & Smale 2002] [Smale & Zhou. 2003] [Q. Wu, et,al 2006][Cucker & Zhou. 2007] [Rosasco et al. 2009] [Christmann & Zhou 2016] [Lin & Rosasco 2016] for more practical discussions (e.g., $0<r<0.5$). This has been carefully discussed in the revised manuscript (see page 5, Section 3.2.1).
>
> **Q2**:	 *How do we know that a $\mathbf{q}$ being a uniform distribution over the most recent observations satisfies Assumption 4. Seems like if $l$ is too large its not likely to hold, but if $l$ is too small then we can't capture any of the non-stationarity.*
>
> **A2**:   Thanks. This assumption is inherited from some recent works, such as [Kuznetsov & Mohri 2020]. Theoretically, [page 16; Kuznetsov & Mohri 2020] have verified that the uniform distribution $\mathbf{q}$ satisfies this assumption, by establishing a sufficient small upper bound of $disc(\mathbf{q})$. This has been carefully discussed in the revised manuscript (see page 8, the first paragraph).
>
> We agree that the parameter $l$ needs to be in an appropriate interval, through tuning in practice, like other hyper-parameters. In this paper, we grid search $l$ by cross validation.
>
> **Q3**:	*Can you provide a bit more color on how Step A/eq (9) is solved? Can you comment on the scalability/space, time complexity of algorithm 1?*
>
> **A3**: The Step A/Eq (9) can be equivalently rewritten as a bilevel optimization problem, as also shown in [Colson et al. 2007][Ochs et al., 2016]:
>
> Outer problem: $\min_{\mathbf{s}} \sum_{t=1}^T (q^*_t − s_t)\ell_\sigma (f_\mathbf{s}(x_t) − y_t) + \lambda_1||\mathbf{q}^* -\mathbf{s}||_2^2 + \lambda_2T^{1/2c}||\mathbf{s}||_2^2$
>
> Inner problem: $f_\mathbf{s} = \arg\max_{f\in \mathcal{H_\mathbf{z}}} \sum_{t=1}^T (q^*_t − s_t)\ell_\sigma (f(x_t) − y_t)$,
>
> where the outer (min) problem parameterized by $\mathbf{s}$, is nested within the inner (max) problem parameterized by $f\in \mathcal{H}_\mathbf{z}$. The outer problem can be solved by standard gradient descent, where in each step, we need to optimize the inner (max) problem with the last updated $\mathbf{s}$. Note that the inner problem is subjected to $f\in \mathcal{H}_\mathbf{z}$. A widely used method for solving this constrained inner problem is DC programming [Tao & An 1998][Kuznetsov&Mohri 2020]. This has been carefully discussed in the revised manuscript (see pages 25-26).
>
> The computational complexity of Algorithm 1 depends on the optimization strategy for DC programming, the training size $T$, dimension $p$ and the iteration times $Z$. We denote by the computational complexity of DC programming $O(DC(T, p))$. Then, the computational complexity of Step A is $O(ZDC(T, p)+ ZT )$ and the computational complexity of Step B is $O(ZT^3p^3)$. Thus, the total computational complexity of Algorithm 1 is $O(ZO(DC) +ZT^3p^3)$. For large scale data, we can further speed up Algorithm 1 by the random Fourier features technique [Rahimi and Recht 2007], which is leaved for future work due to space limitation. This has been carefully discussed in the revised manuscript (see page 26, Remark 3).

---

### Decision · Program_Chairs · 2022-01-20

**Decision:**

Accept (Poster)

**Comment:**

This paper proposes an adaptive sparse Huber additive model for for forecasting non-stationary time series. The prior work has considered similar models for Gaussian innovations which is overly restrictive for a variety of applications such as finance. The results are supported both by theory and experiments. The results are novel and are of interest to ICLR and machine learning communities in general.